# FINE-TUNING QUANTIZED NEURAL NETWORKS WITH ZEROTH-ORDER OPTIMIZATION

**Sifeng Shang**[1]  **Jiayi Zhou**[1]  **Chenyu Lin**[1]  **Minxian Li**[2]  **Kaiyang Zhou**[1,✉]
[1]Hong Kong Baptist University
[2]Nanjing University of Science and Technology
https://github.com/maifoundations/QZO

## ABSTRACT

As the size of large language models grows exponentially, GPU memory has become a bottleneck for adapting these models to downstream tasks. In this paper, we aim to push the limits of memory-efficient training by minimizing memory usage on model weights, gradients, and optimizer states, within a unified framework. Our idea is to eliminate both gradients and optimizer states using zeroth-order optimization, which approximates gradients by perturbing weights during forward passes to identify gradient directions. To minimize memory usage on weights, we employ model quantization, e.g., converting from bfloat16 to int4. However, directly applying zeroth-order optimization to quantized weights is infeasible due to the precision gap between discrete weights and continuous gradients, which would otherwise require de-quantization and re-quantization. To overcome this challenge, we propose Quantized Zeroth-order Optimization (QZO), a simple yet effective approach that perturbs the continuous quantization scale for gradient estimation and uses a directional derivative clipping method to stabilize training. QZO is orthogonal to both scalar-based and codebook-based post-training quantization methods. Compared to full-parameter fine-tuning in 16 bits, QZO can reduce the total memory cost by more than $18\times$ for 4-bit LLMs, and enables fine-tuning Llama-2-13B within a single 24GB GPU.

## 1 INTRODUCTION

Pre-trained large language models (LLMs) (Zhang et al., 2022; Touvron et al., 2023a;b; Grattafiori et al., 2024) have demonstrated great potential in numerous downstream applications, ranging from sentiment classification and text summarization, to more challenging open-ended question answering and creative writing. However, with the model size growing at an exponential rate, adapting LLMs to downstream tasks presents significant challenges to computational resources. For instance, fine-tuning a Llama-7B model stored in bfloat16 typically requires 56GB GPU memory: 14GB for model weights, 14GB for gradients, and another 28GB for optimizer states when adaptive gradient-based optimization methods are used (e.g., the first and second moments in AdamW (Loshchilov & Hutter, 2017), which cost twice the size of gradients). Such an enormous memory cost makes it infeasible for researchers and practitioners with limited computational resources to fine-tune LLMs.

In general, there are four key components that determine memory usage: (1) model weights, (2) gradients (typically the same size as weights), (3) optimizer states (often twice the size as gradients), and (4) activations cached for gradient computation. Since activations are mostly affected by the size of mini-batch, existing memory-efficient training methods mainly target the first three components (Zhao et al., 2024; Malladi et al., 2023). In this work, we aim to push the limits of memory-efficient training by minimizing memory usage on model weights, gradients, and optimizer states, within a *unified* framework.

Our main idea is to eliminate gradients and optimizer states using zeroth-order optimization (Spall, 1992), which gets rid of backpropagation by approximating gradients solely through forward passes (i.e., perturbing model weights to identify gradient directions). When it comes to model weights, the

---

✉ Corresponding author

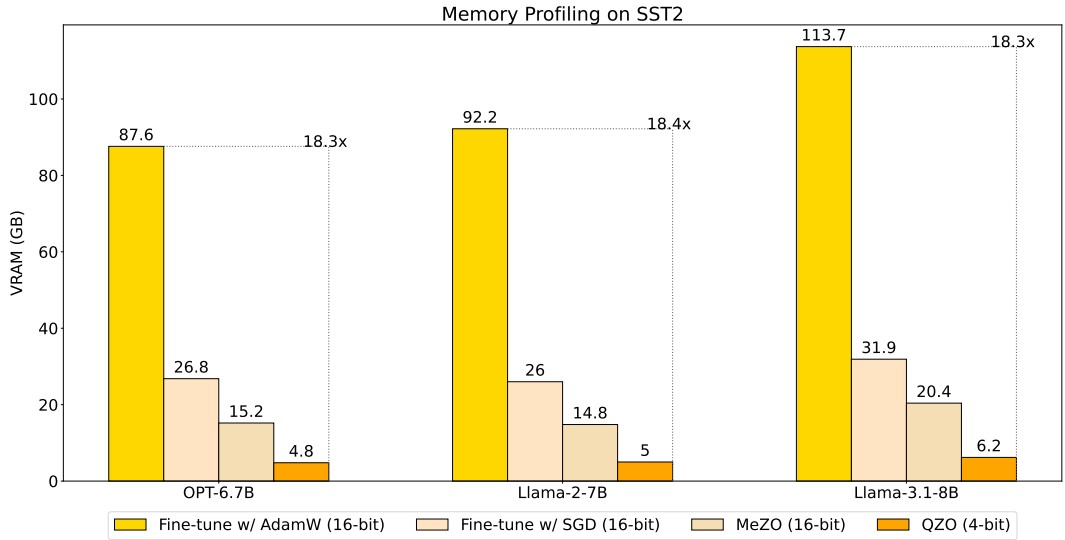

Figure 1: Memory profiling on SST-2 (Socher et al., 2013) with (per-device) batch size set to 1. Fine-tuning w/ AdamW is done with fully-sharded data parallel.

optimal approach is to quantize the weights, e.g., converting from bfloat16 to int4 can significantly cut the memory cost by $4\times$. However, directly applying zeroth-order optimization to quantized weights is non-trivial because (1) quantized weights cannot be perturbed in the continuous space, and (2) the gradients estimated by a zeroth-order optimizer are continuous and therefore cannot be used to update discrete quantized weights (which would otherwise require de-quantization and re-quantization).[1]

To overcome the aforementioned challenges, we propose a novel approach called Quantized Zeroth-order Optimization (QZO), which enables quantized neural networks to be fine-tuned with zeroth-order optimization, hence achieving maximum reduction in memory consumption—compared to full-parameter fine-tuning in 16 bits, QZO significantly reduces the total memory cost by $18\times$ for 4-bit LLMs (see Figure 1). Specifically, QZO approximates the gradients of quantized weights by perturbing the continuous quantization scale parameter(s) rather than the discrete weights, which are kept fixed throughout training. To further stabilize training, we propose a gradient clipping method and provide a theoretical proof to justify that the clipping method essentially reduces the variance of the gradient estimate.

We evaluate QZO on different families of LLMs including OPT (Zhang et al., 2022) and Llama (Touvron et al., 2023b; Grattafiori et al., 2024), as well as using a diverse set of quantization methods. The experiments are conducted on five popular NLP benchmarks including both classification and generation tasks. Using 4-bit LLMs, QZO significantly outperforms both quantized and un-quantized zero-shot models while performing on par with MeZO (Malladi et al., 2023), which applies zeroth-order optimization to un-quantized models. In the extreme quantization case where the model is quantized to 2-bit, QZO still beats the zero-shot baseline by a large margin, demonstrating the effectiveness of QZO in fine-tuning quantized models. We also provide both theoretical evidence and ablation experiments to demonstrate the effectiveness of directional derivative clipping in stabilizing the training, which functions through reducing the variance of the gradient estimates.

## 2   RELATED WORK

**Memory-Efficient Training**   Fine-tuning LLMs often requires a significant amount of GPU memory, making it challenging for model adaptation on resource-constrained hardware. In general, current memory-efficient training methods mainly focus on reducing GPU memory usage for the following components: (1) learnable model weights, (2) gradients, (3) optimizer states storing additional gradient information, and (4) activations cached for gradient computation. To save memory cost

---

[1]By quantization, we refer to post-training quantization throughout this work, unless specified otherwise.

for optimizer states, GaLore (Zhao et al., 2024) projects the first and second moments of gradients in AdamW (Loshchilov & Hutter, 2017) onto a low-rank subspace. MeZO (Malladi et al., 2023) eliminates gradients and optimizer states by using a zeroth-order optimizer (Spall, 1992), which estimates gradients using only forward passes and therefore keeps the memory cost the same as inference. CoLM (Nguyen et al., 2025) uses small mini-batches whose gradients match those of large mini-batches, leading to huge memory reduction in activations. Our approach further pushes the limits of memory-efficient training by fine-tuning quantized LLMs with zeroth-order optimization, which significantly cuts memory usage across all components requiring GPU memory.

**LLM Quantization** Post-training quantization (PTQ) is a popular paradigm for compressing LLMs. Most PTQ methods (Dettmers et al., 2022; Frantar et al., 2023; Lin et al., 2024; Xiao et al., 2023; Ashkboos et al., 2024) reduce the bit width for each model parameter by representing the numerical range with low-precision integers while using full precision for quantization parameters. These methods can achieve up to 4-bit quantization, resulting in up to $4\times$ reduction in memory usage compared to the widely-used BF16 representation. Different from the popular scalar-based quantization paradigm, recent research (Tseng et al., 2024; Egiazarian et al., 2024; Liu et al., 2024) has explored using codebooks for storing full-precision numbers, which are indexed with integers to represent the original model weights. These codebook-based methods can achieve extreme quantization in 2 or 3 bits without observing significant performance drops. Typically, quantized LLMs are not suitable for fine-tuning because continuous gradients cannot be directly applied to updating discrete quantized weights (which would require de-quantization and re-quantization). Our approach seamlessly combines memory-efficient training with quantization to enable fine-tuning on quantized LLMs, achieving maximal reduction on GPU memory usage. More importantly, our approach is orthogonal to most PTQ methods, including both 4-bit and 2-bit quantization methods.

**Zeroth-order Fine-tuning for Quantized Models** Inspired by a foundational approach, ZO-signSGD (Liu et al., 2019), several prior works (Feng et al., 2024; Zhou et al., 2025; Bar & Giryes, 2025) expand on this study to enable the fine-tuning of quantized models, using a shared paradigm that involves quantizing perturbation noises and directly applying sign-based SGD on discrete, quantized weights. Although sharing a similar spirit in minimizing the memory footprint, namely combining zeroth-order optimization with quantization, the proposed QZO approach is inherently more efficient and flexible, as it does not require quantization of perturbation noises or re-quantization of model weights at each optimization iteration. Furthermore, it can be applied to existing scalar-based or codebook-based PTQ methods, such as GPTQ (Frantar et al., 2023) and AQLM (Egiazarian et al., 2024), in a plug-and-play manner.

## 3 METHODOLOGY

### 3.1 BACKGROUND: ZEROTH-ORDER OPTIMIZATION

Zeroth-order optimization (ZO) methods are often used in cases where gradients and higher-order derivatives of the objective cannot be directly computed or are unreliable (Conn et al., 2009). The pioneering work, Simultaneous Perturbation Stochastic Approximation (SPSA) (Spall, 1992), is defined as follows,

**Definition 3.1** (Simultaneous Perturbation Stochastic Approximation, SPSA (Spall, 1992)). *Given a model parameterized by $\boldsymbol{\theta} \in \mathbb{R}^d$ and a loss function $\mathcal{L}$, SPSA estimates the gradients of $\boldsymbol{\theta}$ on a mini-batch $\mathcal{B}$ using the following formula:*

$$\hat{\nabla}_{\boldsymbol{\theta}} \mathcal{L}(\boldsymbol{\theta}; \mathcal{B}) = \frac{\mathcal{L}(\boldsymbol{\theta} + \epsilon \boldsymbol{z}; \mathcal{B}) - \mathcal{L}(\boldsymbol{\theta} - \epsilon \boldsymbol{z}; \mathcal{B})}{2\epsilon} \boldsymbol{z} \approx \boldsymbol{z}\boldsymbol{z}^{\top} \nabla_{\boldsymbol{\theta}} \mathcal{L}(\boldsymbol{\theta}; \mathcal{B}), \tag{1}$$

*where $\boldsymbol{z} \in \mathbb{R}^d$ is a random vector sampled from $\mathcal{N}(0, \boldsymbol{I}_d)$, and $\epsilon$ the perturbation scale.*

Built on top of SPSA, a recent work (Malladi et al., 2023) proposed memory-efficient zeroth-order optimization (MeZO) for LLMs. In particular, MeZO uses random seeds as a trick to eliminate the storage cost of $\boldsymbol{z}$, and as a result, the memory footprint is kept the same level as inference. MeZO also replaces the regular SGD (Robbins & Monro, 1951) with zeroth-order stochastic gradient descent (ZO-SGD), which is defined below:

**Definition 3.2** (Zeroth-Order Stochastic Gradient Descent, ZO-SGD (Malladi et al., 2023)). *Given a learning rate $\eta$, ZO-SGD updates the parameters $\boldsymbol{\theta}_t$ at $t$-th step using gradients estimated by SPSA as follows:*

$$\boldsymbol{\theta}_{t+1} = \boldsymbol{\theta}_t - \eta \hat{\nabla}_{\boldsymbol{\theta}_t} \mathcal{L}(\boldsymbol{\theta}_t; \mathcal{B}_t) \tag{2}$$

*where $\mathcal{B}_t$ denotes the input mini-batch at step $t$.*

### 3.2 QZO: QUANTIZED ZEROTH-ORDER OPTIMIZATION

QZO minimizes the memory usage not only on gradients and optimizer states but also on model weights—this can save huge memory cost when using large models of more than 10B parameters, e.g., when using bfloat16, a 10B model's weights consume 20GB of memory, while using int4, the weights only take 5GB of memory. QZO consists of two core modules: Quantized Simultaneous Perturbation Stochastic Approximation (Q-SPSA), and directional derivative clipping. The former extends SPSA to quantized weights while the latter stabilizes training by reducing the variance of gradient estimation.

#### 3.2.1 FROM SPSA TO Q-SPSA

SPSA (Eq. 1) cannot be directly applied to quantized weights because (1) quantized weights are discrete and therefore cannot be perturbed in the continuous space, and (2) the continuous gradients cannot be used to update discrete weights, which would otherwise require de-quantization and re-quantization. To overcome these challenges, we propose Quantized Simultaneous Perturbation Stochastic Approximation (Q-SPSA), which only applies perturbation to the continuous quantization scale. We begin by introducing quantization and de-quantization, which are two essential steps in model quantization. Concretely, for each single element $w$ in a weight set $\mathcal{W}$, these two steps can be formulated as

$$\overline{w} = \lfloor \frac{w}{\Delta} \rceil, \tag{3}$$

$$w = \Delta \cdot \overline{w}, \tag{4}$$

where $\Delta$ denotes an element-wise quantization scale, and $\overline{w}$ the quantized counterpart stored using lower bits. The weight set $\mathcal{W}$ is determined by the choice of quantization group, while the implementation of $\Delta$ varies among different quantization methods. For example, when $\Delta = \frac{\text{absmax}(\mathcal{W})}{2^{k-1}-1}$, Eqs. 3 and 4 refer to the standard scalar-based quantization in $k$-bit.

Since the de-quantization process in Eq. 4 aligns with the normal forward propagation, we decompose the model parameters $\boldsymbol{\theta}$ in Eq. 1 into $\boldsymbol{\Delta} \odot \bar{\boldsymbol{\theta}}$, and perturb the scaling component $\boldsymbol{\Delta}$ while keeping the discrete weights $\bar{\boldsymbol{\theta}}$ fixed. Therefore, Q-SPSA can be formulated as

**Definition 3.3** (Quantized Simultaneous Perturbation Stochastic Approximation, Q-SPSA). *Given a quantized model with integer parameters $\bar{\boldsymbol{\theta}} \in \mathbb{R}^d$ and quantization scales $\boldsymbol{\Delta}$, and a loss function $\mathcal{L}$, Q-SPSA estimates the gradients of $\boldsymbol{\Delta}$ over a mini-batch $\mathcal{B}$ using the following formula:*

$$\hat{\nabla}_{\boldsymbol{\Delta}} \mathcal{L}(\boldsymbol{\Delta} \odot \bar{\boldsymbol{\theta}}; \mathcal{B}) = \frac{\mathcal{L}((\boldsymbol{\Delta} + \epsilon \boldsymbol{z}) \odot \bar{\boldsymbol{\theta}}; \mathcal{B}) - \mathcal{L}((\boldsymbol{\Delta} - \epsilon \boldsymbol{z}) \odot \bar{\boldsymbol{\theta}}; \mathcal{B})}{2\epsilon} \boldsymbol{z}$$
$$\approx \boldsymbol{z}\boldsymbol{z}^\top \nabla_{\boldsymbol{\Delta}} \mathcal{L}(\boldsymbol{\Delta} \odot \bar{\boldsymbol{\theta}}; \mathcal{B}), \tag{5}$$

*where $\boldsymbol{z} \in \mathbb{R}^d$ is a random vector sampled from $\mathcal{N}(0, \boldsymbol{I}_d)$, $\epsilon$ the perturbation scale, and $\odot$ the Hadamard product.*

Similar to MeZO, all quantization scales within a linear layer are perturbed to save computation. In practice, one may choose to fine-tune the continuous quantization scale only, or combine Q-SPSA with SPSA to jointly update the unquantized counterparts. It is worth noting that Q-SPSA can be applied to both scalar-based and codebook-based quantization methods: in the experiments we show that our approach can successfully fine-tune both 4-bit LLMs quantized by the scalar-based GPTQ (Frantar et al., 2023) and 2-bit LLMs quantized by the codebook-based AQLM (Egiazarian et al., 2024) (in this case both the channel-wise scales and un-quantized weights are updated).

### 3.2.2 DDC: DIRECTIONAL DERIVATIVE CLIPPING

Gradient estimation via ZO is notorious for causing unstable training due to large gradient variance (Malladi et al., 2023). This was also observed when combining Q-SPSA with the vanilla ZO-SGD method in our preliminary experiments where training often collapsed. To mitigate this problem, we propose Directional Derivative Clipping (DDC) and apply this method before updating the model with ZO-SGD at each optimization step.

Specifically, the gradient estimate in Eq. 5 can be viewed as a product of the random vector $z$ and the estimated directional derivative of loss function along $z$ w.r.t. $\Delta$ (which is essentially a scalar). Let $d$ denote the estimated directional derivative, Eq. 5 can be re-written as $\hat{\nabla}_\Delta \mathcal{L}(\Delta \odot \bar{\theta}; \mathcal{B}) = d \cdot z$. Then, DDC applies clipping to $d$ by:

$$d' = \begin{cases} C, & \text{if } d > C \\ d, & d \in [-C, C] \\ -C, & \text{if } d < -C \end{cases} \tag{6}$$

where $C$ is a non-negative constant. The gradient estimate then becomes $\hat{\nabla}_\Delta \mathcal{L}'(\Delta \odot \bar{\theta}; \mathcal{B}) = d' \cdot z$, which is plugged into ZO-SGD. We provide theoretical evidence to highlight that DDC can reduce the variance of the gradient estimate and thereby stabilize the training. We first propose the following theorem as a preliminary to our analysis. The proof of Theorem 1 is available in Appendix A.

**Theorem 1.** *Clipped gradient estimate $\hat{\nabla}_\Delta \mathcal{L}'(\Delta \odot \bar{\theta}; \mathcal{B})$ is an unbiased estimate of the full gradient of loss w.r.t quantization sclaes $\nabla_\Delta \mathcal{L}(\Delta \odot \bar{\theta})$.*

Since $d'^2 \leq d^2$ by definition of DDC in Eq. 6, the following inequality holds:

$$\mathbb{E}[||\hat{\nabla}_\Delta \mathcal{L}'(\Delta \odot \bar{\theta}; \mathcal{B})||^2] = \mathbb{E}[d'^2 ||z||^2] \leq \mathbb{E}[d^2 ||z||^2] = \mathbb{E}[||\hat{\nabla}_\Delta \mathcal{L}(\Delta \odot \bar{\theta}; \mathcal{B})||^2] \tag{7}$$

Therefore, the element-wise variance of the clipped gradient estimate has the following derivation:

$$\begin{aligned} Var[\hat{\nabla}_{\Delta_k} \mathcal{L}'(\Delta \odot \bar{\theta}; \mathcal{B})] &= \mathbb{E}[||\hat{\nabla}_{\Delta_k} \mathcal{L}'(\Delta \odot \bar{\theta}; \mathcal{B})||^2] - \mathbb{E}[\hat{\nabla}_{\Delta_k} \mathcal{L}'(\Delta \odot \bar{\theta}; \mathcal{B})]^2 \\ &\leq \mathbb{E}[||\hat{\nabla}_{\Delta_k} \mathcal{L}(\Delta \odot \bar{\theta}; \mathcal{B})||^2] - \mathbb{E}[\hat{\nabla}_{\Delta_k} \mathcal{L}'(\Delta \odot \bar{\theta}; \mathcal{B})]^2 \\ &= Var[\hat{\nabla}_{\Delta_k} \mathcal{L}(\Delta \odot \bar{\theta}; \mathcal{B})] + \mathbb{E}[\hat{\nabla}_{\Delta_k} \mathcal{L}(\Delta \odot \bar{\theta}; \mathcal{B})]^2 - \mathbb{E}[\hat{\nabla}_{\Delta_k} \mathcal{L}'(\Delta \odot \bar{\theta}; \mathcal{B})]^2 \\ &= Var[\hat{\nabla}_{\Delta_k} \mathcal{L}(\Delta \odot \bar{\theta}; \mathcal{B})] + \left(\nabla_{\Delta_k} \mathcal{L}(\Delta \odot \bar{\theta})\right)^2 - \mathbb{E}[\hat{\nabla}_{\Delta_k} \mathcal{L}'(\Delta \odot \bar{\theta}; \mathcal{B})]^2 \end{aligned} \tag{8}$$

By Theorem 1, $Var[\hat{\nabla}_{\Delta_k} \mathcal{L}'(\Delta \odot \bar{\theta}; \mathcal{B})] \leq Var[\hat{\nabla}_{\Delta_k} \mathcal{L}(\Delta \odot \bar{\theta}; \mathcal{B})]$ holds almost surely.

Our experimental results in Section 4.3 also reveal that DDC effectively stabilizes the training through rectifying abnormal loss values, and the ablation study also demonstrates that QZO is relatively robust to the magnitude of $C$.

### 3.2.3 ALGORITHM

We summarize QZO in Algorithm 1. Note that although the quantization scales are perturbed per parameter in the pseudo code, in practice one may perturb the entire quantization scales of a linear layer to save training time (Malladi et al., 2023).

**Remarks** QZO seamlessly combines ZO with quantization and therefore leads to maximum reduction in memory usage: gradients and optimizer states are eliminated while model weights are compressed. To further cut memory usage on activations, one can divide the batch size while increasing the total number of optimization steps, or release activations during forward passes since ZO does not need to cache activations for gradient computation.

## 4 EXPERIMENTS

### 4.1 EXPERIMENTAL SETUP

**Models and Datasets** We evaluate our approach using three 7B-level LLMs, namely OPT-6.7B (Zhang et al., 2022), Llama-2-7B (Touvron et al., 2023b), and Llama-3.1-

---

**Algorithm 1** Quantized Zeroth-order Optimization

---

**Require:** quantization scales $\mathbf{\Delta} \in \mathbb{R}^d$, quantized weights $\bar{\boldsymbol{\theta}} \in \mathbb{R}^d$, loss function $\mathcal{L} : \mathbb{R}^d \to \mathbb{R}$ learning rate $\eta_t$, optimization steps $T$, perturbation scales $\epsilon$, clipping threshold $C$.

> **for** $t = 1...T$ **do**
>> Sample batch of inputs $\mathcal{B}$ and random seed $s$
>> $\mathbf{\Delta} \leftarrow \text{PERTURB\_SCALES}(\mathbf{\Delta}, \epsilon, s)$
>> $\ell_+ \leftarrow \mathcal{L}(\mathbf{\Delta} \odot \bar{\boldsymbol{\theta}}; \mathcal{B})$                  $\triangleright$ `1`st` forward pass`
>> $\mathbf{\Delta} \leftarrow \text{PERTURB\_SCALES}(\mathbf{\Delta}, -2\epsilon, s)$
>> $\ell_- \leftarrow \mathcal{L}(\mathbf{\Delta} \odot \bar{\boldsymbol{\theta}}; \mathcal{B})$                  $\triangleright$ `2`nd` forward pass`
>> $\mathbf{\Delta} \leftarrow \text{PERTURB\_SCALES}(\mathbf{\Delta}, \epsilon, s)$
>> $d \leftarrow (\ell_+ - \ell_-)/(2\epsilon)$
>> $d' \leftarrow \text{CLIP}(d, -C, C)$      $\triangleright$ `Directional derivative clipping, Eq. 6`
>> Reset random number generator with seed $s$
>> **for** $\Delta_i \in \mathbf{\Delta}$ **do**
>>> $z \sim \mathcal{N}(0, 1)$
>>> $\Delta_i \leftarrow \max(\Delta_i - \eta_t * d' * z, 0)$          $\triangleright$ `Ensure non-negative scales`
>> **end for**
> **end for**
>
> **procedure** PERTURB\_SCALES$(\mathbf{\Delta}, \epsilon, s)$
>> Reset random number generator with seed $s$
>> **for** $\Delta_i \in \mathbf{\Delta}$ **do**
>>> $z \sim \mathcal{N}(0, 1)$
>>> $\Delta_i \leftarrow \Delta_i + \epsilon z$
>> **end for**
> **end procedure**

---

8B (Grattafiori et al., 2024), and one large-sized model with 13B parameters, i.e., Llama-2-13B (Touvron et al., 2023b). For QZO, the 7B models are quantized to 4-bit while the 13B model to 2-bit to test QZO's effectiveness under extreme quantization. Following prior work (Malladi et al., 2023), we evaluate our approach on five popular NLP datasets covering both classification and generation tasks. Specifically, for classification, we use SST2 (Socher et al., 2013) and three subsets from SuperGLUE collection (Wang et al., 2019), i.e., RTE (Dagan et al., 2005; Haim et al., 2006; Giampiccolo et al., 2007; Bentivogli et al., 2009), CB (De Marneffe et al., 2019) and BoolQ (Clark et al., 2019). For generation, we use SQuAD (Rajpurkar et al., 2016), which is a question answering dataset. Following the common practice, we randomly sample 1,000 examples for training, 500 examples for validation, and 1,000 examples for testing. We report accuracy for classification tasks, whereas the metric for generation tasks is F1 score.

**Baseline Methods** A wide range of baseline methods is chosen for comparison to justify QZO's effectiveness. Specifically, QZO is compared with: (1) Zero-Shot, and Zero-Shot-Q, the original and quantized zero-shot models, respectively, which are viewed as the lower-bound; (2) Fine-tuning on 16-bit models, which is considered as the upper-bound;[2] (3) MeZO (Malladi et al., 2023), which applies ZO to un-quantized models.

**Implementation Details** For 4-bit quantization, we apply GPTQ (Frantar et al., 2023) to the 7B-level LLMs (i.e., OPT-6.7B, Llama-2-7B, and Llama-3.1-8B).[3] The quantization group in GPTQ is set to 128. For extreme quantization in 2-bit, we apply AQLM (Egiazarian et al., 2024) with 1 codebook of 16 bits to Llama-2-13B.[4] We use QZO to fine-tune the channel-wise scales in AQLM. Following prior work (Egiazarian et al., 2024; Tseng et al., 2024), the un-quantized parts are jointly fine-tuned using the regular SPSA and ZO-SGD. To accelerate QZO fine-tuning in 2-bit, we also

---

[2]Due to limited budget on computational resources, fine-tuning experiments are conducted with SGD optimizer unless otherwise specified.

[3]https://github.com/ModelCloud/GPTQModel

[4]https://huggingface.co/ISTA-DASLab/Llama-2-7b-AQLM-2Bit-1x16-hf

Table 1: Experiments based on OPT-6.7B, Llama-2-7B, and Llama-3.1-8B. Zero-Shot and Zero-Shot-Q serve as the lower-bound, while Fine-tuning (with SGD) is the upper-bound. QZO works well across different model architectures on all datasets, with a memory footprint significantly lower than MeZO and Fine-tuning.

| | | Model Precision | Memory Profiling | Classficiation | | | | Generation |
| | | | | SST-2 | RTE | CB | BoolQ | SQuAD |
|---|---|---|---|---|---|---|---|---|
| OPT-6.7B | Fine-tuning | 16 bits | 26.8GB | 95.4 | 79.8 | 73.2 | 69.6 | 77.6 |
| | Zero-Shot | 16 bits | - | 61.2 | 55.2 | 51.8 | 59.5 | 36.5 |
| | Zero-Shot-Q | 4 bits | - | 60.1 | 53.8 | 51.8 | 59.1 | 35.9 |
| | MeZO | 16 bits | 14.8GB | 93.0 | 64.6 | 67.9 | 66.8 | 79.6 |
| | QZO | 4 bits | 4.8GB | 87.6 | 61.7 | 67.9 | 66.4 | 78.5 |
| Llama-2-7B | Fine-tuning | 16 bits | 26.0GB | 92.8 | 63.2 | 60.7 | 75.0 | 83.7 |
| | Zero-Shot | 16 bits | - | 58.1 | 61.7 | 32.1 | 66.0 | 55.6 |
| | Zero-Shot-Q | 4 bits | - | 58.5 | 53.4 | 35.7 | 64.6 | 53.6 |
| | MeZO | 16 bits | 14.8GB | 83.5 | 58.1 | 67.9 | 69.6 | 80.7 |
| | QZO | 4 bits | 5.0GB | 90.0 | 59.2 | 69.6 | 68.2 | 85.5 |
| Llama-3-8B | Fine-tuning | 16 bits | 31.9GB | 93.7 | 71.5 | 62.5 | 83.4 | 84.9 |
| | Zero-Shot | 16 bits | - | 59.6 | 45.8 | 46.4 | 66.1 | 64.8 |
| | Zero-Shot-Q | 4 bits | - | 58.7 | 50.2 | 37.5 | 65.0 | 59.2 |
| | MeZO | 16 bits | 20.5GB | 92.5 | 70.0 | 91.1 | 83.4 | 86.9 |
| | QZO | 4 bits | 6.3GB | 93.0 | 66.8 | 69.6 | 78.2 | 88.3 |

Table 2: Training statistics collected on SST-2. Overall, QZO is both memory-efficient and computation-efficient.

| | | Trainable Paramters | Total FLOPs (SST-2) |
|---|---|---|---|
| OPT-6.7B | Fine-tuning | $6.65 \times 10^9$ | $2.17 \times 10^{16}$ |
| | MeZO | $6.65 \times 10^9$ | $9.91 \times 10^{17}$ |
| | QZO | $5.03 \times 10^7$ | $8.19 \times 10^{13}$ |
| Llama-2-7B | Fine-tuning | $6.74 \times 10^9$ | $2.47 \times 10^{16}$ |
| | MeZO | $6.74 \times 10^9$ | $1.13 \times 10^{18}$ |
| | QZO | $5.06 \times 10^7$ | $2.26 \times 10^{16}$ |
| Llama-3.1-8B | Fine-tuning | $8.03 \times 10^9$ | $2.48 \times 10^{16}$ |
| | MeZO | $8.03 \times 10^9$ | $1.13 \times 10^{18}$ |
| | QZO | $5.45 \times 10^7$ | $7.9 \times 10^{16}$ |

modify AQLM's `Triton` inference kernel to disentangle matrix reconstruction and matrix-vector multiplication.[5] For QZO, we set the learning rate to $10^{-7}$, the batch size to 16, training steps to 20k, the perturbation scale $\epsilon$ to $10^{-3}$, and the clipping threshold $C$ to 100. For fine-tuning experiments with SGD, the learning rate is initialized as $8 \times 10^{-4}$ with a linearly scheduled decay, and the batch size is set to 8. A single Nvidia RTX 4090 GPU (24GB) is used for all experiments (except Fine-tuning, which requires an A100 80GB GPU). For MeZO, we adopt the official code.[6]

## 4.2 MAIN RESULTS

**QZO on 4-bit Quantization** Table 1 compares QZO with different baselines across three model architectures on the five NLP datasets. The detailed training statistics are shown in Table 2. Following MeZO, memory profiling measures the peak memory usage during the first 100 optimization steps. The dataset used for memory profiling is SST2 and the (per-device) batch size is set to 1 to test the minimum VRAM requirement. We summarize our main findings below.

---

[5]https://github.com/triton-lang/triton
[6]https://github.com/princeton-nlp/MeZO

Table 3: Experiments based on Llama-2-13B. QZO demonstrates strong potential under extreme quantization.

| | | Model Precision | Memory Profiling | Classification | | | | Generation |
|---|---|---|---|---|---|---|---|---|
| | | | | SST-2 | RTE | CB | BoolQ | SQuAD |
| Llama-2-13B | Zero-Shot-Q | 2 bits | - | 57.6 | 53.1 | 46.4 | 69.2 | 55.4 |
| | QZO | 2 bits | 5.78GB | 80.5 | 54.5 | 55.4 | 70.2 | 59.4 |

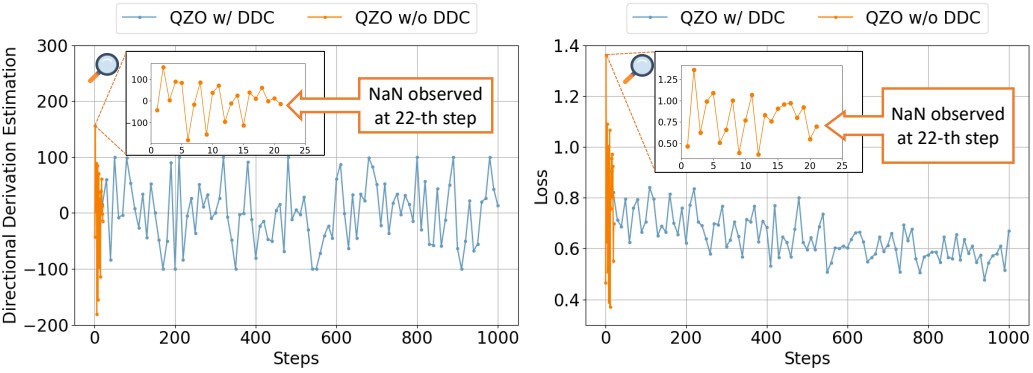

Figure 2: Directional derivatives (left) and loss values (right) collected during the early 1,000 training steps. Without DDC, the training is extremely unstable, often leading to abnormal directional derivatives and eventually NaN values for the loss.

*QZO demonstrates effectiveness consistently across all model architectures and NLP tasks.* Specifically, QZO achieves significant improvements over Zero-Shot-Q, meaning that QZO successfully fine-tunes these quantized LLMs. On most datasets, QZO performs on par with MeZO, despite using *3× less memory*; sometimes QZO even beats MeZO with noticeable margins, e.g., 85.5 vs. 80.7 on SQuAD when using Llama-2-7B. It is worth highlighting that MeZO is based on 16-bit models while QZO is based on 4-bit models *with much lower precision*. Compared with the upper-bound, i.e., fine-tuning, the gap is still huge on some of the tasks. This makes sense because ZO methods rely merely on forward passes for gradient estimation, which would be much less accurate than that of backpropagation.

*QZO demonstrates both memory-efficiency and computation-efficiency.* QZO pushes memory-efficiency to the extreme by eliminating gradients and optimizer states while reducing weights precision. Therefore, the memory usage is minimal compared to the baselines like MeZO and Fine-tuning. Table 2 compares QZO with MeZO and Fine-tuning on learnable parameter count and FLOPs. It is worth noting that QZO uses only about 1% of the trainable parameters and 1% of the FLOPs of MeZO. This is because QZO only fine-tunes the continuous quantization scale while leaving most weights (which are quantized) fixed. We expect the difference to be further increased when more powerful quantization methods are used.

**QZO on 2-bit Quantization** Table 3 shows that QZO beats the zero-shot model with significant margins. The results strongly justify QZO's effectiveness under extreme quantization. QZO has the potential to be applied to on-device learning scenarios for edge devices.

### 4.3 ABLATION STUDIES

In this section, we mainly evaluate the DDC component. Recall that DDC (Directional Derivative Clipping, Eq. 6) clips abnormal directional derivatives estimated via QZO (i.e., $d$ in Eq. 6). We use QZO to train two Llama-2-7B models, with and without using DDC, and record the directional derivatives and loss values for the first 1,000 steps. Figure 2 shows that without DDC the directional

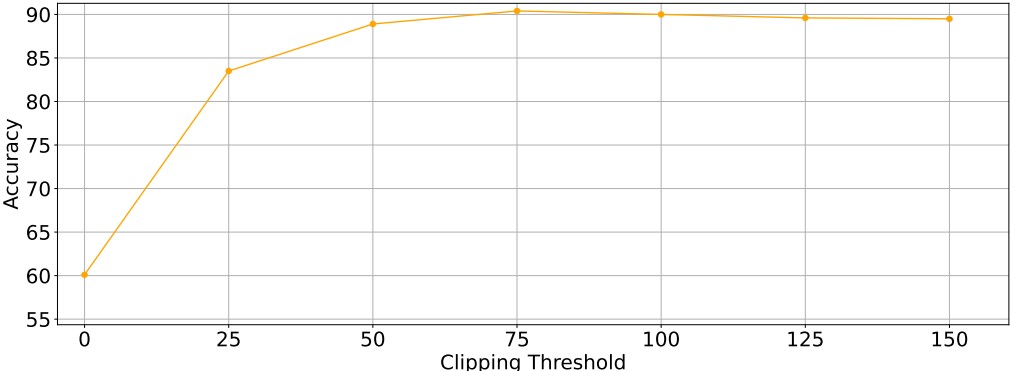

Figure 3: Impact of clipping threshold. A small $C$ effectively avoids abnormal directional derivatives, but suffers from underfitting due to a small optimization step size. A large $C$ fixes this issue, but may introduce the risk of producing abnormal values. Note the performance at $C = 0$ corresponds with zero-shot accuracy of the quantized model.

derivative often gets abnormal values that go beyond the range of $[-C, C]$ ($C$ is the clipping threshold in Eq. 6), leading to NaN value for the loss (which means the training collapses).

We also study how sensitive QZO is to the clipping threshold $C$. Intuitively, a small $C$ should effectively avoid abnormal directional derivatives, but may suffer from underfitting due to a small optimization step size. A large $C$ fixes this issue, but also increases the risk of producing abnormal values. For quantitative analysis, we train Llama-2-7B models on SST-2 with different values of $C$ and record the final accuracies. The results are presented in Figure 3. The trend of the line plot suggests underfitting at $C \leq 50$, and stable performances can be observed when $C \geq 75$. When $C$ is set to a value bigger than 150, the training becomes unstable and sometimes collapse, which algins with the observation in Figure 2 (QZO w/ DDC can be seen as setting $C$ to an infinitely large value).

## 5 CONCLUSION, LIMITATIONS, AND FUTURE WORK

QZO enables fine-tuning quantized neural networks via ZO, which greatly reduces memory usage related to model weights, gradients, and optimizer states. We show that QZO works for a wide range of LLMs and is compatible with both scalar-based and codebook-based quantization methods. When using 4-bit LLMs, QZO achieves performance on par with MeZO, while using $3\times$ less GPU memory. In the extreme quantization scenario, QZO successfully fine-tunes 2-bit LLama-2-13B across different NLP datasets. The results indicate that QZO has the potential to be applied to on-device learning for edge devices.

In addition to LLMs, we have also applied QZO to fine-tuning text-to-image generation models, namely Stable Diffusion 3.5 Large (Esser et al., 2024). The results and discussions are presented in Appendix F. QZO fine-tunes Stable Diffusion 3.5 Large using only 12.4GB of memory in a single Nvidia RTX 4090 GPU. The visualization results are also encouraging: the data distribution generated by QZO is visually closer to the ground truth than the zero-shot model.

However, QZO has some limitations. First, QZO's performance depends on how good the quantization method is. Specifically, if the quantization method has a large quantization error, this makes the forward passes in ZO noisy and therefore could make the gradient estimation less accurate. On the other hand, QZO could benefit from a better quantization method with higher accuracy. Therefore, practitioners are suggested to choose high-precision quantization methods for QZO to maximize the gains.

Second, the performance on diffusion models lags behind LLMs because there is a noticeable gap between QZO's images and the ground truth. This may be caused by the mismatch in the noise scheduling between ZO and diffusion. One potential solution is to redesign the noise scheduling in ZO such that it aligns with diffusion. We leave this as future work.

## 6 ETHICS STATEMENT

We clarify that our research is free from the issues in the code of ethics. Our research focuses on the efficiency of LLM training and does not include any human subjects. The datasets used do not include sensitive content that violates data privacy.

## 7 REPRODUCIBILITY STATEMENT

Our code has been publicly released to ensure reproducibility of experiments. All the datasets involved are also publicly accessible. The proof of Theorem 1 is provided in the Appendix.

## 8 ACKNOWLEDGEMENT

This research is supported by Hong Kong Research Grants Council Early Career Scheme (No. 22200824).

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

# A PROOF OF THEOREM 1

## A.1 PRELIMINARY FORMULATIONS

In the SGD algorithm, the stochastic gradient of a mini-batch $\mathcal{B}$ is given by

$$\nabla_{\boldsymbol{\theta}}\mathcal{L}(\boldsymbol{\theta};\mathcal{B}) = \frac{1}{|\mathcal{B}|}\sum_{k\in\mathcal{B}}\nabla\ell_{\boldsymbol{\theta}}(\boldsymbol{\theta};x_k)$$

In our QZO, following Definition 3.3, the estimated gradient is formulated as

$$\hat{\nabla}_{\boldsymbol{\Delta}}\mathcal{L}(\boldsymbol{\Delta}\odot\bar{\boldsymbol{\theta}};\mathcal{B}) = d\boldsymbol{z} \approx \boldsymbol{z}\boldsymbol{z}^{\top}\nabla_{\boldsymbol{\Delta}}\mathcal{L}(\boldsymbol{\Delta}\odot\bar{\boldsymbol{\theta}};\mathcal{B}) \qquad \boldsymbol{z}\sim\mathcal{N}(0,\boldsymbol{I}_d)$$

where $d = \frac{\mathcal{L}((\boldsymbol{\Delta}+\epsilon\boldsymbol{z})\odot\bar{\boldsymbol{\theta}};\mathcal{B})-\mathcal{L}((\boldsymbol{\Delta}-\epsilon\boldsymbol{z})\odot\bar{\boldsymbol{\theta}};\mathcal{B})}{2\epsilon}\boldsymbol{z}$. Note that $\nabla_{\boldsymbol{\theta}}\mathcal{L}(\boldsymbol{\theta};\mathcal{B})$ and $\hat{\nabla}_{\boldsymbol{\Delta}}\mathcal{L}(\boldsymbol{\Delta}\odot\bar{\boldsymbol{\theta}};\mathcal{B})$ are unbiased estimate of the full gradient $\nabla_{\boldsymbol{\theta}}\mathcal{L}(\boldsymbol{\theta})$ and full gradient of loss w.r.t quantization scales $\nabla_{\boldsymbol{\Delta}}\mathcal{L}(\boldsymbol{\Delta}\odot\bar{\boldsymbol{\theta}})$, respectively.

## A.2 PROOF

In this section, we detail the proof of Theorem 1. Let $C$ be the clipping threshold, the clipped gradient estimate can be reformulated by:

$$\hat{\nabla}_{\boldsymbol{\Delta}}\mathcal{L}'(\boldsymbol{\Delta}\odot\bar{\boldsymbol{\theta}};\mathcal{B}) = \text{clip}(d,-C,C)\boldsymbol{z} = d'\boldsymbol{z} \tag{9}$$

*Proof.* Suppose that the mini-batch $\mathcal{B}$ is sampled from the dataset $\mathcal{D}$, and $||\mathcal{D}||$ denotes the number of mini-batches in the dataset.

$$\mathbb{E}_{\mathcal{B},\boldsymbol{z}}[\hat{\nabla}_{\boldsymbol{\Delta}}\mathcal{L}'(\boldsymbol{\Delta}\odot\bar{\boldsymbol{\theta}};\mathcal{B})] = \mathbb{E}_{\boldsymbol{z}}[\frac{1}{||\mathcal{D}||}\sum_{i\in\mathcal{D}}d_i'\boldsymbol{z}]$$

$$= \mathbb{E}_{\boldsymbol{z}}[\frac{1}{N}\sum_{i\in\mathcal{D},d_i<|C|}d_i'\boldsymbol{z} + \frac{1}{M}\sum_{i\in\mathcal{D},d_i>|C|}d_i'\boldsymbol{z}]$$

$$= \mathbb{E}_{\boldsymbol{z}}[\frac{1}{N}\sum_{i\in\mathcal{D},d_i<|C|}d\boldsymbol{z} + \frac{1}{M}\sum_{i\in\mathcal{D},d_i>|C|}|C|\boldsymbol{z}] \tag{10}$$

$$= \mathbb{E}_{\boldsymbol{z}}[\frac{1}{N}\sum_{i\in\mathcal{D},d_i<|C|}\boldsymbol{z}\boldsymbol{z}^{\top}\nabla_{\boldsymbol{\Delta}}\mathcal{L}(\boldsymbol{\Delta}\odot\bar{\boldsymbol{\theta}};\mathcal{B})] + \mathbb{E}_{\boldsymbol{z}}[\frac{|C|}{M}\sum_{i\in\mathcal{D},d_i>|C|}\boldsymbol{z}] \tag{11}$$

$$= \mathbb{E}_{\boldsymbol{z}}[\mu_{i\in\mathcal{D},d_i<|C|}(\boldsymbol{z}\boldsymbol{z}^{\top}\nabla_{\boldsymbol{\Delta}}\mathcal{L}(\boldsymbol{\Delta}\odot\bar{\boldsymbol{\theta}};\mathcal{B}))] + 0 \tag{12}$$

$$= \mathbb{E}_{\boldsymbol{z}}[\boldsymbol{z}\boldsymbol{z}^{\top}]\mathbb{E}_{\mathbb{B}}[\nabla_{\boldsymbol{\Delta}}\mathcal{L}(\boldsymbol{\Delta}\odot\bar{\boldsymbol{\theta}};\mathcal{B})] \tag{13}$$

$$= \nabla_{\boldsymbol{\Delta}}\mathcal{L}(\boldsymbol{\Delta}\odot\bar{\boldsymbol{\theta}})$$

Eq. 10 equals Eq. 11 as $\epsilon \to 0$. In Eq. 12, $\mu$ represents the sample mean of the $N$ observations, and the transition from Eq. 12 to Eq. 13 holds because the sample mean $\mu$ is an unbiased estimate of the expectation. $\square$

# B LOSS-ACCURACY CURVES

We include the plots of loss-accuracy curves in Figure 4 to visually compare the convergence and training stability of QZO and the zeroth-order baseline, MeZO. Specifically, we use QZO and MeZO to fine-tune an OPT-6.7B model on the SST-2 task. We report loss values every 10 steps and test set accuracy every 4,000 steps. The illustrated loss and accuracy curves for QZO show a clear convergence pattern similar to that of MeZO, indicating its training stability as a zeroth-order optimization method.

# C IMPACT ON TRAINING SET SAMPLING

To explore the impact of training set sampling, we fine-tune OPT-6.7B models with QZO on downstream NLP tasks with three different seeds, which control the training set partition. The results of

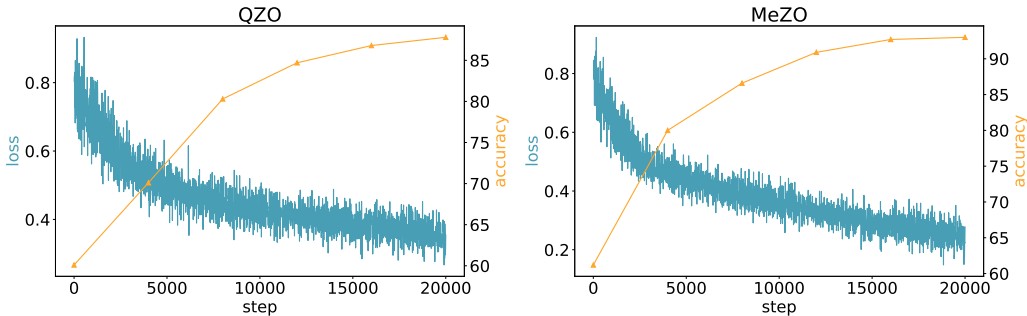

Figure 4: Plots for loss-accuracy curves. We fine-tune an OPT-6.7B model using QZO and MeZO on SST-2. The loss values are reported every 10 steps, while the test accuracy is recorded every 4,000 steps.

Table 4: Experiments with three different seeds controlling the training set partition. The average results are reported with error bars representing the 95% confidence intervals.

|  | Seed | SST-2 | RTE | CB | BoolQ | SQuAD |
|---|---|---|---|---|---|---|
| OPT-6.7B | 0 | 87.6 | 61.7 | 67.9 | 66.4 | 78.5 |
|  | 1 | 88.2 | 59.2 | 71.4 | 65.9 | 77.2 |
|  | 2 | 89.2 | 62.1 | 69.6 | 64.8 | 76.3 |
|  | Average | $88.3 \pm 0.9$ | $61.0 \pm 1.8$ | $69.6 \pm 2.0$ | $65.7 \pm 0.9$ | $77.3 \pm 1.3$ |

each run, together with the averages and corresponding 95% confidence intervals, are presented in the Table 4. The results demonstrate that QZO is robust to training set sampling, as the performance of different seeds does not exhibit a significant discrepancy.

## D    DISCUSSION WITH PEFT METHODS

We first clarify that the proposed QZO is orthogonal to parameter-efficient fine-tuning (PEFT) methods, as it directly tunes the scales of quantized models without requiring any additional trainable adapters. This also indicates that the QZO and PEFT methods can be combined to fine-tune quantized models jointly using the zeroth-order optimizer. For a quantitative study, we conduct a series of experiments to compare the performance of QZO, QLoRA (Dettmers et al., 2023), and the combination of both. The results are presented in Table 5. Specifically, we use the methods listed in the table to fine-tune a (quantized) OPT-6.7B model. The low-rank adapters are injected into the weight matrices of the q_proj and v_proj layers across all experiments conducted with PEFT methods. And the LoRA rank and LoRA alpha are set to 8 and 16 consistently. We also naively combine QZO with zeroth-order QLoRA to create the variant QZO+QLoRA by fine-tuning low-rank adapters during the first half of training with zeroth-order QLoRA, while training the remaining quantization scales in layers without LoRA using QZO in the second half. Based on the results, we report the following findings:

**(i) First-order methods consistently outperform zeroth-order methods.**    This is reasonable, since real gradients are used for optimization rather than approximations. We emphasize that although QLoRA fine-tuning shows low memory usage in memory profiling, this efficiency is achieved only when gradient checkpointing is enabled, and a paged optimizer is used. In comparison, QZO and its variant directly achieve memory efficiency through zeroth-order optimization without bells and whistles.

**(ii) It is possible to combine QZO and QLoRA to improve the performance while keeping the memory efficiency.**    Based on the results, the QZO+QLoRA variant could generally outperform the original QZO while maintaining a similar low memory footprint. We also emphasize that the QZO+QLoRA variant requires only $8,000$ steps, saving $60\%$ of the total fine-tuning steps required

Table 5: Comparison with parameter-efficient fine-tuning methods. The LoRA rank and LoRA alpha are set to 8 and 16 consistently across all experiments. QZO+QLoRA is a variant by fine-tuning low-rank adapters during the first half of training with *zeroth-order* QLoRA, while training the remaining quantization scales in layers without LoRA using QZO in the second half.

|  | Method | Model Precision | Memory Profiling | SST-2 | RTE | CB |
|---|---|---|---|---|---|---|
| First-order | Fine-tuning | 16 bits | 26.8 GB | 95.4 | 79.8 | 73.2 |
|  | LoRA | 16 bits | 14.0 GB | 95.6 | 83.8 | 71.4 |
|  | QLoRA | 4 bits | 5.6 GB | 96.1 | 84.1 | 67.9 |
| Zeroth-order | MeZO | 16 bits | 14.8 GB | 93.0 | 64.6 | 67.9 |
|  | QZO | 4 bits | 4.8 GB | 87.6 | 61.7 | 67.9 |
|  | QZO+QLoRA | 4 bits | 4.9 GB | 93.3 | 61.7 | 69.9 |

by the original QZO and MeZO (Malladi et al., 2023). We believe that combining QZO and PEFT methods can be more effectively accomplished than our naive approach, suggesting a promising direction for future research.

## E  CONVERGENCE GUARANTEE AND TRAINING TIME

A theoretical study from prior work (Malladi et al., 2023) indicates that the zeroth-order optimizer guarantees a convergence rate similar to SGD, with a slowdown factor proportional to the local effective rank of the Hessian of loss. Although QZO perturbs the quantization scales rather than the model weights in full precision, we believe this finding for the zeroth-order optimizer also supports our method. We also note that QZO requires less training time than MeZO (Malladi et al., 2023), as inference kernels can accelerate the forward pass for quantized models. For example, when fine-tuning an OPT-6.7B on SST-2 with a single NVIDIA 4090, MeZO training takes approximately 4 hours and 26 minutes, whereas QZO takes approximately 2 hours and 16 minutes.

## F  RESULTS WITH STABLE DIFFUSION

### F.1  EXPERIMENTS ON STABLE DIFFUSION

**Model and Dataset**  We evaluate our approach on Stable Diffusion 3.5 Large (Esser et al., 2024), the current state-of-the-art text-to-image model (the largest among the 3.5 series). We choose the Styled Image Dataset (Ganjdanesh et al., 2024) for evaluation, which includes the Frosting Lane, PS1, Tarot, and Yarn styles, with 10,000 image-caption pairs per style. For each style, the images of $512 \times 512$ resolution are split using the 8:2 train-test ratio.

**Implementation Details**  For QZO, NF4 quantization is applied to Stable Diffusion 3.5 Large using BitsAndBytes (Dettmers et al., 2023). The batch size is set to 16. The learning rate is set to 1e-6. The perturbation scale $\epsilon$ is set to 1e-3. The total number of training steps is 20k. Following the common practice, only the DiT part in Stable Diffusion 3.5 Large is fine-tuned.

**Memory Usage**  Stable Diffusion 3.5 Large consists of a VAE, a DiT, and three text encoders (CLIP-ViT/G, CLIP-ViT/L, and T5-XXL). For regular training in fp16/bf16, this model requires 0.37GB for the VAE, 21.26GB for the text encoders, 16.2GB for the DiT, 16.2GB for gradients, and 32.4GB for optimizer states, totaling 86.43GB of memory usage (without even considering other overheads like caches and buffers). In contrast, QZO takes only 12.4GB of memory for fine-tuning, which can easily fit into a single Nvidia RTX 4090 GPU (24GB). To our knowledge, this is the first work showing that fine-tuning Stable Diffusion 3.5 Large can be done on a single consumer-grade GPU.

**Qualitative Results and Discussion**  The results are visualized in Figure 5 to Figure 8. Overall, the results are encouraging: the data distribution generated by QZO is visually closer to the ground truth than the zero-shot model, which suggests that QZO works to some extent for fine-tuning quantized

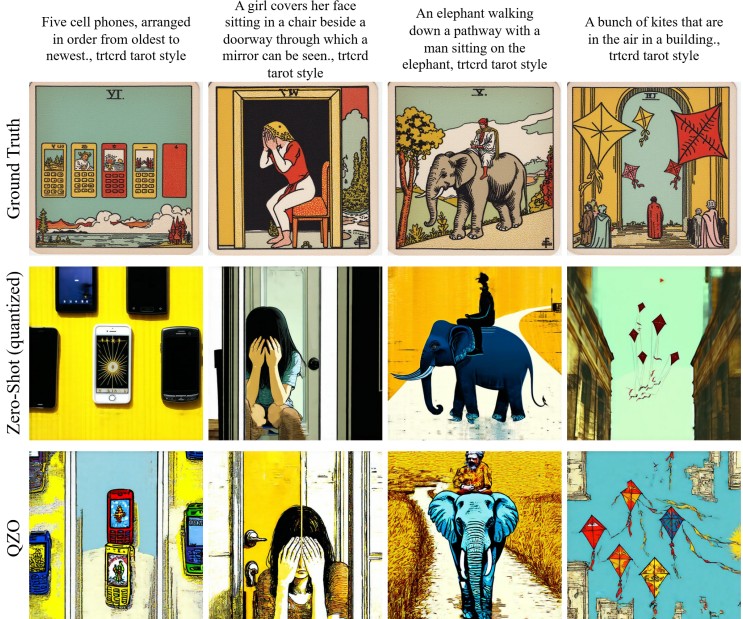

Figure 5: Tarot style image generation results.

text-to-image models. However, there is still a noticeable gap between QZO's images and the ground truths.

We discuss two reasons that may explain why QZO does not produce the same level of performance as in LLMs. First, unlike discrete probability modeling as in LLMs, Stable Diffusion is essentially a regression model that predicts continuous noise values. This architectural difference leads to a sensitivity issue: the deviations of the estimated gradients via ZO are directly manifested as differences at the pixel level during image generation, and such errors are propagated through continuous output, resulting in fidelity degradation.

Second, recall that ZO introduces noise perturbations in latent representations. Consider a linear layer without bias, $y = Wa$, the forward call in QZO leads to $y = (W + \eta d \cdot z)a$ after one optimization step, where $\eta$ denotes the learning rate and $d$ the estimated directional derivative discussed in Eq. 6. This update injects additional Gaussian noise $\eta d \cdot z$ into the activations, which is propagated through the denoising process and thus disrupts the pre-configured noise schedule (it acts as conflicting noise patterns). The diffusion model is unable to simultaneously remove the scheduled and ZO-induced noise, thus resulting in incomplete denoising.

# G  THE USE OF LARGE LANGUAGE MODELS (LLMs)

We restrict our use of LLMs to grammar checking and writing polishing. Content translation is not used throughout the paper, and any significant use that could lead to research misconduct is intentionally avoided.

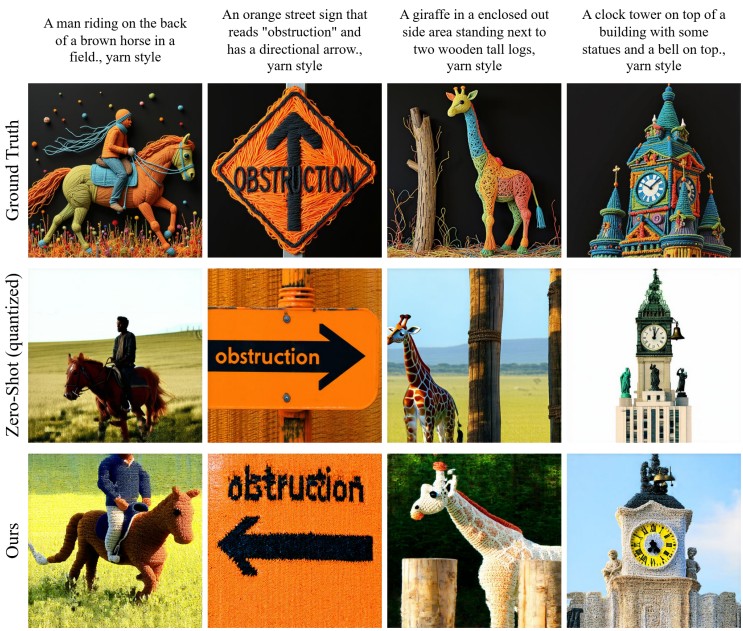

Figure 6: Yarn style generation comparison.

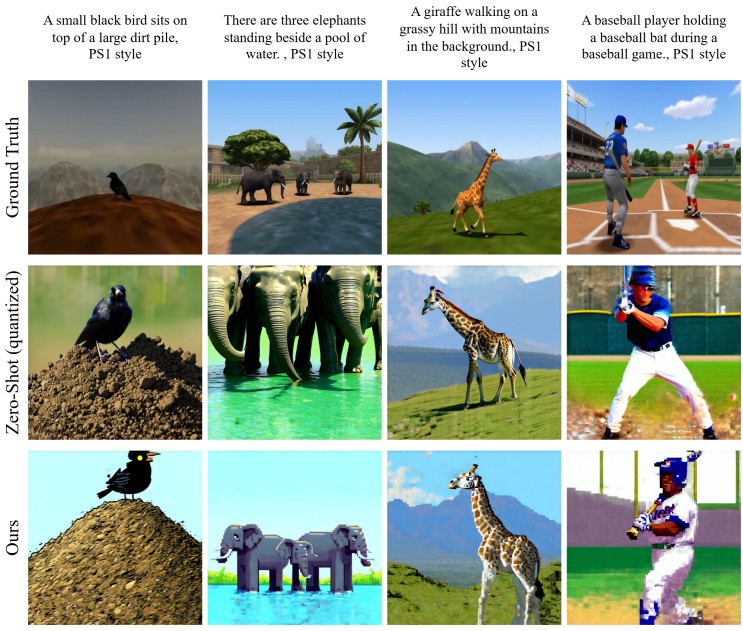

Figure 7: PS1 style generation comparison.

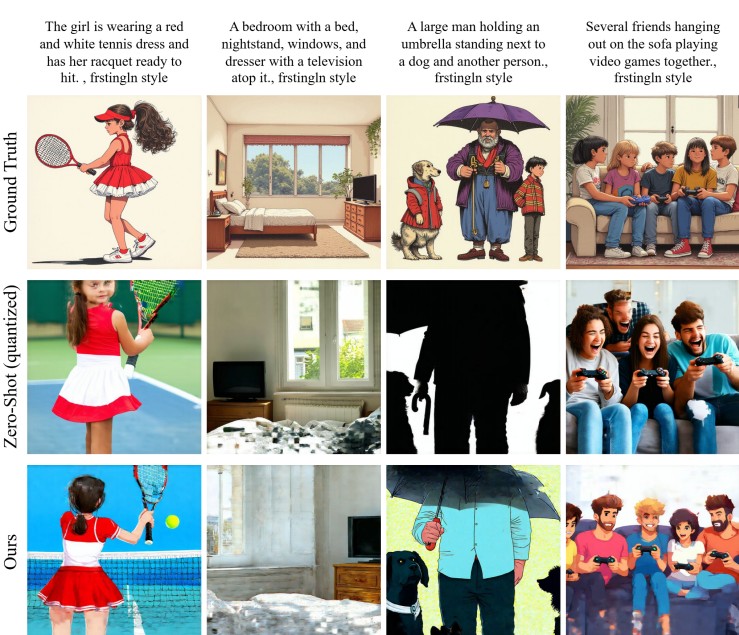

Figure 8: Frosting Lane style generation comparison.

