# OpenReview forum: "Fine-tuning Quantized Neural Networks with Zeroth-order Optimization"
_ICLR.cc/2026/Conference — ICLR 2026 Poster_

### Official Review · Reviewer_mqYe · 2025-10-22

**Soundness:** 3
**Presentation:** 3
**Contribution:** 3
**Rating:** 6
**Confidence:** 3

**Summary:**

The large model size of LLM poses a challenge to computational resources. This work aims to reduce the memory cost of all the weights, gradients and optimizer states through one algorithm. The authors propose an algorithm QZO. Specifically, it uses the quantized model to reduce the memory of weights, and proposes a novel technique Q-SPSA that only perturbs the continuous quantization scale to compute the zeroth-order gradient with low memory cost. In addition, QZO adopts the clipping on the obtained zeroth-order gradient to increase the stability of training. A theoretical analysis is provided to verify its effectiveness.

In the experiments on several tasks, benefiting from reducing the memory cost of weights, gradients and optimizer states, QZO significantly reduces the memory usage compared to other baselines. Meanwhile, the gap of performances between it and full fine-tuning is not large, even higher than full fine-tuning in some cases. The ablation study shows the effects of gradient clipping.

The main contribution of this work is that it proposes an algorithm that could update the quantized weight with the zeroth-order gradient. This idea is verified in the experiments: the memory cost is around 1/3 of that of the SOTA algorithm.

**Strengths:**

Previous works either reduce the memory of weights by quantization, or reduce the memory in the optimization step by the zo gradient. This work designs the Q-SPSA method to apply the zo gradient on the quantized weights. The originality and novelty are good. The experimental results reveal that this is an effective method for memory-efficient LLM training, indicating that this is a significant work. The motivation of this work is clear, and the description for the algorithm is easy to follow.

**Weaknesses:**

This work does not provide any convergence guarantee for the proposed algorithm. The experiments focus on fine-tuning the model on downstream tasks and do not include any pretraining tasks.

Typo: the parenthesis in equation (3) seems to be incorrect.

**Questions:**

The experimental results in Table 1 show that QZO always outperforms baselines in the generalization task, but is comparable to or even obviously worse than baselines in the classification tasks. Could you give some explanations for this, i.e. why QZO performs so well in the generalization task?

Table 2 shows that the trainable parameters of QZO are significantly less than baselines. The authors explain that “This is because QZO only fine-tunes the continuous quantization scale”. I think QZO just reduces the precision of the trainable parameters, further reducing the flops. But the number of trainable parameters should be unchanged, as each element $w$ has a corresponding $\Delta$ to compute its zo gradient. Could you respond to this issue?

---

> ### Author Response · Authors · 2025-11-20
> **Response to Reviewer mqYe**
>
> We appreciate your valuable suggestions and positive comments! We feel really motivated that you found our novelty good, our method effective, and our motivation clear. Below are our responses to further address your concerns.
>
> > This work does not provide any convergence guarantee for the proposed algorithm. The experiments focus on fine-tuning the model on downstream tasks and do not include any pretraining tasks.
>
> A theoretical study from prior work [1] indicates that the zeroth-order optimizer guarantees a convergence rate similar to SGD, with a slowdown factor proportional to the local effective rank of the Hessian of loss. Although QZO perturbs the quantization scales rather than the model weights in full precision, we believe this finding for the zeroth-order optimizer also applies to our method.
>
> The primary focus of this work is fine-tuning LLMs in a memory-efficient manner, and we acknowledge that the effectiveness of QZO for pre-training remains underexplored. Compared to fine-tuning, pre-training starts from a randomly initialized model and uses a different optimization objective. In this case, some of the assumptions for zeroth-order fine-tuning may no longer hold (e.g., the Hessian's small effective rank, as mentioned above). We might expand QZO to support memory-efficient pre-training in our future work.
>
> > Typo: the parenthesis in equation (3) seems to be incorrect.
>
> The parenthesis in Eq. 3 (i.e., $\lfloor \cdot \rceil$) denotes a round-to-nearest operator, which is adopted in the formulations of scalar-based quantization methods. We will also review our manuscript thoroughly to identify any potential typos.
>
> > The experimental results in Table 1 show that QZO always outperforms baselines in the generalization task, but is comparable to or even obviously worse than baselines in the classification tasks. Could you give some explanations for this, i.e. why QZO performs so well in the generalization task?
>
> I think there is a typo in the comment: it should be 'generation' task rather than 'generalization' task. Based on the results presented in Table 1, both QZO and MeZO generally perform better than the SGD fine-tuning baseline on the generation task, suggesting this may be a common strength for zeroth-order optimization methods.
>
> > Table 2 shows that the trainable parameters of QZO are significantly less than baselines. The authors explain that “This is because QZO only fine-tunes the continuous quantization scale”. I think QZO just reduces the precision of the trainable parameters, further reducing the flops. But the number of trainable parameters should be unchanged, as each element $w$ has a corresponding $\Delta$ to compute its zo gradient. Could you respond to this issue?
>
> The experiments on QZO presented in Table 2 are conducted on LLMs quantized using GPTQ (also known as OPTQ [2]), a group-wise quantization method. This indicates that for each layer, weight elements in a quantization group share the same scale. Since the quantization group size is 128 in our experiments, the trainable parameters are also reduced to about 1% compared to the full-parameter fine-tuning.
>
> We thank you again for the feedback and hope our responses address your concerns!

---

> ### Author Response · Authors · 2025-11-28
>
> Hello! It's been a few days since our initial discussion post. I wonder if our responses have addressed your concerns. If not, please feel free to start another round of discussion. We welcome any valuable suggestions from you to help improve our work! We would also be very grateful if you could raise the final score once your concerns have been addressed. Looking forward to your reply!

---

### Official Review · Reviewer_p1QC · 2025-10-29

**Soundness:** 2
**Presentation:** 3
**Contribution:** 2
**Rating:** 2
**Confidence:** 4

**Summary:**

The paper studies memory-efficient fine-tuning of LLMs using quantization and zeroth-order optimization. As the size of LLMs grows quickly, the memory cost of fine-tuning becomes expensive. Zeroth-order optimization has shown significant improvement in saving memory for LLMs fine-tuning. The paper further studies how to even reduce it by quantization and proposes quantized zeroth-order optimization (QZO). Theoretical analysis is provided to show that the proposed method reduces variance in gradient estimation. Empirical experiments on fine-tuning LLMs on downstream tasks show great savings in fine-tuning memory, while preserving similar performance.

**Strengths:**

The paper is clearly written and has nice structure. The main claims are verified with either theory or experiments. The studied topic is well-motivated, and the problem setting is interesting and relevant to the community. By designing memory-efficent methods for LLMs fine-tuning, more and more people that do not necessarily have large computation resources can benefit from the rapid development and advancements of LLMs.

**Weaknesses:**

I believe there is a large body of related works that are not properly discussed, especially on fine-tuning with zeroth-order methods and quantization.

1. The study of quantization for zeroth-order fine-tuning of LLMs is not new; see [1,2,3]. The paper should properly discuss what are the major differences and contributions compared to these related works. It is hard to evaluate the novelty and contributions of this work as such discussions are missing. Indeed, the settings and algorithms in this work seem to be very similar to these previous papers. Therefore, I am confused and thus require additional explanations from the authors.

2. There are also other ways to further save memory compared to MeZO, such as sparse MeZO [4,5]. This could also serve as a valid baseline to compare memory consumption and model performance.

3. This is less relevant. The clipped method used in the algorithm is also popular and introduced in the context of private and zeroth-order fine-tuning; see [6,7].

The experiments are a bit limited.

1. The paper considers OPT and Llama, with model size 7B, 8B, and 13B. It is not clear whether the method also works for other model family, and whether the algorithm scales to even larger models, as fine-tuning 13B model does not really reach the memory bottleneck.

2. Comparisons with LoRA and QLoRA are missing. LoRA and QLoRA with gradient checkpoint and gradient accumulation should also significantly reduce the memory while keeping similar performance. I am not sure how quantized zeroth-order optimization compares to that.

3. The considered downstreams tasks are a bit limited. Only 5 different tasks are used.

[1] ZOQO: Zero-Order Quantized Optimization. ICASSP 2025. (arXiv:2501.06736)

[2] QuZO: Quantized Zeroth-Order Fine-Tuning for Large Language Models. 2025. (arXiv:2502.12346)

[3] Stepping Forward on the Last Mile. NeurIPS 2024. (arXiv:2411.04036)

[4] Sparse MeZO: Less Parameters for Better Performance in Zeroth-Order LLM Fine-Tuning. 2024. (arXiv: 2402.15751)

[5] Zeroth-Order Fine-Tuning of LLMs with Extreme Sparsity. 2024. (arXiv:2406.02913)

[6] DPZero: Private Fine-Tuning of Language Models without Backpropagation. ICML 2024. (arXiv:2310.09639)

[7] Private Fine-tuning of Large Language Models with Zeroth-order Optimization. TMLR 2024. (arXiv:2401.04343)

**Questions:**

Questions are already mentioned in weaknesses.

---

> ### Author Response · Authors · 2025-11-20
> **Response to Reviewer p1QC - Page 1 / 2**
>
> Thanks for your thoughtful feedback and constructive suggestions! We are motivated that you found our paper clearly written, our study well motivated, and the problem setting interesting and relevant. We make the following responses to address your concerns.
>
> > I believe there is a large body of related works that are not properly discussed, especially on fine-tuning with zeroth-order methods and quantization.
>
> Thank you for the supplement of related works. We provide a brief discussion below to address your concerns.
>
> > The study of quantization for zeroth-order fine-tuning of LLMs is not new; see [1,2,3]. The paper should properly discuss what are the major differences and contributions compared to these related works. It is hard to evaluate the novelty and contributions of this work as such discussions are missing. Indeed, the settings and algorithms in this work seem to be very similar to these previous papers. Therefore, I am confused and thus require additional explanations from the authors.
>
> We clarify that although these related works share a similar setting with ours, i.e., fine-tuning quantized models with zeroth-order optimization, their algorithmic paradigms differ from QZO. The methods in [1,2,3] perturb the quantized weights with quantized noise based on Zero-Sign Stochastic Gradient Descent (ZO-SignSGD[7]). In contrast, QZO perturbs the quantization scales rather than quantized weights and directly utilizes the ZO-SGD[9] optimizer to update the scales. As a result, QZO does not require requantization during optimization steps and could potentially be combined with the algorithms in [1,2,3] to jointly fine-tune both the quantized weights and scales.
>
> > There are also other ways to further save memory compared to MeZO, such as sparse MeZO [4,5]. This could also serve as a valid baseline to compare memory consumption and model performance.
>
> The methods presented in [4,5] integrate sparsity with zeroth-order optimization, thereby achieving a speedup in training and better performance. Based on our understanding, the Sparse-MeZO proposed in [4] cannot improve memory efficiency beyond the original MeZO, as it applies unstructured sparsity to the perturbation noises, rather than the model weights. The method proposed in [5] reduces the memory footprint via post-training quantization and leverages sparsity to select a subset of trainable parameters from the quantized model. From our perspective, the design of sparsity-based parameter selection is orthogonal to QZO, as our method enables fine-tuning of the quantized model, which remains untouched in [5], and thereby the two can be jointly combined for better performance.
>
> > This is less relevant. The clipped method used in the algorithm is also popular and introduced in the context of private and zeroth-order fine-tuning; see [6,7].
>
> Thank you for providing the missing related works! Our clipping method differs slightly from those in [6,7], as we clip the estimates of directional derivatives. We have also provided a theoretical proof to support its effectiveness. We will include these related works in our revision and compare their methodological differences.
>
> > The experiments are a bit limited.
>
> > The paper considers OPT and Llama, with model size 7B, 8B, and 13B. It is not clear whether the method also works for other model family, and whether the algorithm scales to even larger models, as fine-tuning 13B model does not really reach the memory bottleneck.
>
> We consider fine-tuning LLMs on consumer-grade GPUs, such as a single NVIDIA RTX 4090 with only 24GB of GPU memory, or a device with even less. In this case, fine-tuning of 7B-level models has already reached the memory bottleneck using regular first-order methods, such as SGD or AdamW. In light of this, we believe our experiments have sufficiently demonstrated the memory efficiency of QZO, as it minimizes the memory footprints to less than 8GB.

---

> > ### Author Response · Authors · 2025-11-20
> > **Response to Reviewer p1QC - Page 2 / 2**
> >
> > > Comparisons with LoRA and QLoRA are missing. LoRA and QLoRA with gradient checkpoint and gradient accumulation should also significantly reduce the memory while keeping similar performance. I am not sure how quantized zeroth-order optimization compares to that.
> >
> > Thanks for the valuable suggestions. We clarify that QZO is also orthogonal to PEFT methods, as it does not require additional parameters and directly tunes the scales of a quantized model. This also indicates that QZO can be combined with PEFT methods in zeroth-order optimization to enhance performance while maintaining memory efficiency. For a quantitative study, we conducted a series of experiments to fine-tune an OPT-6.7B model with QZO and PEFT methods, and the results are presented in the table below. Specifically, for all the PEFT-related experiments, the low-rank adapters are injected into the `q_proj` and `v_proj` layers, and the LoRA rank and LoRA alpha are set to $8$ and $16$ consistently. We also naively combine QZO with zeroth-order QLoRA to create the variant 'QZO+QLoRA' by fine-tuning low-rank adapters during the first half of training with zeroth-order QLoRA, while training the remaining quantization scales in layers without LoRA using QZO in the second half.
> >
> > Based on the results, we report the following findings:
> >
> > 1. First-order methods consistently outperform zeroth-order methods. This is reasonable, since real gradients are used for optimization rather than approximations. We emphasize that although QLoRA fine-tuning shows low memory usage in memory profiling, this efficiency is achieved only when gradient checkpointing is enabled and a paged optimizer is used. In comparison, QZO and its variant directly achieve memory efficiency through zeroth-order optimization without bells and whistles.
> > 2. We highlight that it is possible to combine QZO and QLoRA to improve the performance while keeping the memory efficiency. Based on the results, the QZO+QLoRA variant could generally outperform the original QZO while maintaining a similar low memory footprint. We also emphasize that the QZO+QLoRA variant requires only $8,000$ steps, saving $60\%$ of the total fine-tuning steps required by the original QZO and MeZO. We believe that combining QZO and PEFT methods can be more effectively accomplished than our naive approach, suggesting a promising direction for future research.
> >
> > |                  | Methods     | Model Precision | Memory Profiling | **SST-2** | **RTE** | **CB** |
> > | ---------------- | ----------- | --------------- | ---------------- | --------- | ------- | ------ |
> > | **First-order**  | Fine-tuning | 16 bits         | 26.8 GB          | 95.4      | 79.8    | 73.2   |
> > |                  | LoRA        | 16 bits         | 14.0 GB          | 95.6      | 83.8    | 71.4   |
> > |                  | QLoRA       | 4 bits          | 5.6 GB           | 96.1      | 84.1    | 67.9   |
> > | **Zeroth-order** | MeZO        | 16 bits         | 14.8 GB          | 93.0      | 64.6    | 67.9   |
> > |                  | QZO         | 4 bits          | 4.8 GB           | 87.6      | 61.7    | 67.9   |
> > |                  | QZO+QLoRA   | 4 bits          | 4.9 GB           | 93.3      | 61.7    | 69.6   |
> >
> > > The considered downstreams tasks are a bit limited. Only 5 different tasks are used.
> >
> > Thank you for the suggestion! The downstream tasks involved already span a wide range of applications, including sentiment classification, entailment recognition, natural language reference, question answering, and language generation. We may consider supplementing more downstream tasks in our revision.
> >
> > We thank you again for the feedback and hope our responses address your concerns!

---

> > > ### Author Response · Authors · 2025-11-20
> > > **Response to Reviewer p1QC - References**
> > >
> > > References
> > >
> > > [1] Bar N, Giryes R. ZOQO: Zero-Order Quantized Optimization[C]//ICASSP 2025-2025 IEEE International Conference on Acoustics, Speech and Signal Processing (ICASSP). IEEE, 2025: 1-5.
> > >
> > > [2] Zhou J, Yang Y, Zhen K, et al. QuZO: Quantized zeroth-order fine-tuning for large language models[J]. arXiv preprint arXiv:2502.12346, 2025.
> > >
> > > [3] Feng C, Zhuo S, Zhang X, et al. Stepping forward on the last mile[J]. Advances in Neural Information Processing Systems, 2024, 37: 94851-94870.
> > >
> > > [4] Liu Y, Zhu Z, Gong C, et al. Sparse mezo: Less parameters for better performance in zeroth-order llm fine-tuning[J]. arXiv preprint arXiv:2402.15751, 2024.
> > >
> > > [5] Guo W, Long J, Zeng Y, et al. Zeroth-Order Fine-Tuning of LLMs with Extreme Sparsity[C]//2nd Workshop on Advancing Neural Network Training: Computational Efficiency, Scalability, and Resource Optimization (WANT@ ICML 2024).
> > >
> > > [6] Zhang L, Li B, Thekumparampil K K, et al. DPZero: Private Fine-Tuning of Language Models without Backpropagation[C]//International Conference on Machine Learning. PMLR, 2024: 59210-59246.
> > >
> > > [7] Tang X, Panda A, Nasr M, et al. Private Fine-tuning of Large Language Models with Zeroth-order Optimization[J]. Transactions on Machine Learning Research.
> > >
> > > [8] Liu S, Chen P Y, Chen X, et al. signSGD via zeroth-order oracle[C]//International conference on learning representations. 2019.
> > >
> > > [9] Malladi S, Gao T, Nichani E, et al. Fine-tuning language models with just forward passes[J]. Advances in Neural Information Processing Systems, 2023, 36: 53038-53075.

---

> ### Comment · Reviewer_p1QC · 2025-11-27
>
> Thanks so much for your response!
>
> My largest concern is on the comparion with existing works [1,2,3]. I see no reasons why these works were ignored. They are the most relevant works compared to all the other papers in the reference and can be easily found by a proper literature review. The current explanation from the authors is a bit too brief. Although I understand now what are the differences, a more detailed discussion in the revision will be necessary to consider acceptance of the paper.
>
> Is it possible for the authors to provide a revision on how these works [1,2,3] will be referred? In my opinion, the extensive discussions should at least include
>
> 1. Given these existing methods, why is it still necessary to introduce a new algorithm? Is it the case that these methods have limitations and how does the new algorithm address these limitations?
>
> 2. What are the novelty and benefits of the new algorithm? Which part in the design of the new method brings this benefit? Is it possible to provide empirical experiments to justify the benefits of the new algorithm over exising methods?
>
> Feel free to add/remove. It is also ok to provide empirical comparisons later considering the time.
>
> All my other concerns have been addressed. I will increase my score to 4.

---

> > ### Author Response · Authors · 2025-11-29
> >
> > Thank you for the valuable feedback! Regarding your primary concern about the related works [1,2,3], we have added a new paragraph to the 'Related Works' section in our updated rebuttal revision for your review. We highlight two unique advantages of QZO that improve both the efficiency and flexibility in contrast to the previous works: (1) QZO is free from quantization of perturbation noises or re-quantization model weights during each optimization step, and (2) QZO can be adapted to existing post-training quantization methods in a plug-and-play manner. We also apologize that due to a lack of source code for all the proposed methods [1,2,3], we are unable to provide the empirical comparisons during the discussion period. We hope our response could further address your concerns!

---

### Official Review · Reviewer_g1TF · 2025-10-30

**Soundness:** 3
**Presentation:** 2
**Contribution:** 2
**Rating:** 4
**Confidence:** 4

**Summary:**

This paper introduces Quantized Zeroth-Order Optimization (QZO), a straightforward yet powerful method that estimates gradients by perturbing the continuous quantization scale and enhances training stability through a directional derivative clipping technique.

**Strengths:**

1. This work proposes Quantized Zeroth-order Optimization (QZO), which significantly reduces training memory usage.
2. The optimization of quantized weights requires no de-quantization or re-quantization.
3. The memory optimization is effective — for example, fine-tuning Llama-2-13B on a single 24GB GPU.

**Weaknesses:**

1. In the Introduction, the paper lacks a clear distinction between full-parameter fine-tuning and parameter-efficient fine-tuning. The current discussion implicitly assumes full-parameter fine-tuning, which is insufficient.
2. The experiments are also inadequate. Comparisons with parameter-efficient fine-tuning methods (e.g., LoRA, QLoRA) are missing, both in terms of memory consumption and fine-tuning performance. As a result, the paper does not demonstrate the trade-off between performance and efficiency, nor does it convincingly show the practical utility of the proposed method.
3. As shown in Table 1, the proposed method suffers from a notable performance drop during training.

**Questions:**

0. Please first review the items listed under Weakness.
1. Is there a comparison with the Adam optimizer? The gap between the zero-shot optimizer and Adam remains unclear.
3. In addition to the final evaluation results, providing training loss curves would help to more intuitively compare training convergence and stability.

---

> ### Author Response · Authors · 2025-11-20
> **Response to Reviewer g1TF - Page 1 / 2**
>
> Thank you for the feedback and suggestions! We appreciate your positive comments about our method design and its effectiveness in memory efficiency. We make the following responses to further address your concerns.
>
> > In the Introduction, the paper lacks a clear distinction between full-parameter fine-tuning and parameter-efficient fine-tuning. The current discussion implicitly assumes full-parameter fine-tuning, which is insufficient.
>
> Thank you for the valuable suggestion! We provide a brief discussion about our work and parameter-efficient fine-tuning (PEFT) to address your concerns. PEFT methods freeze the original model weights during fine-tuning, while injecting a small amount of trainable adapters on top. A representative study of PEFT is the low-rank adapters (LoRA[1]), which assume that the gradient during fine-tuning exhibits a low-rank property, and thereby use trainable low-rank matrices as adapters. A recent work, QLoRA[2], enhances LoRA's efficiency by keeping the base model quantized. While QLoRA shares a similar setting with our work and has a low memory footprint, we emphasize that it requires first-order optimization, and its memory efficiency is achieved when gradient checkpointing and the paged optimizer are activated. In comparison, QZO is well-suited to zeroth-order optimization, where gradient computation is prohibited. It also achieves minimal memory cost without bells and whistles. In terms of method design, QZO is also orthogonal to PEFT methods, as it does not require additional parameters and directly tunes the scales of a quantized model. In light of this, QZO can be combined with PEFT methods in zeroth-order optimization to enhance performance while maintaining memory efficiency.
>
> > The experiments are also inadequate. Comparisons with parameter-efficient fine-tuning methods (e.g., LoRA, QLoRA) are missing, both in terms of memory consumption and fine-tuning performance. As a result, the paper does not demonstrate the trade-off between performance and efficiency, nor does it convincingly show the practical utility of the proposed method.
>
> Thank you for the suggestion! We have conducted a series of experiments to investigate the performance of QZO and PEFT methods, and the results are presented in the table below. Specifically, we use the involved methods to fine-tune a (quantized) OPT-6.7B model. Across all PEFT-related experiments, the low-rank adapters are injected into the `q_proj` and `v_proj` layers, and the LoRA rank and LoRA alpha are set to $8$ and $16$ consistently. We also naively combine QZO with zeroth-order QLoRA to create the variant 'QZO+QLoRA' by fine-tuning low-rank adapters during the first half of training with zeroth-order QLoRA, while training the remaining quantization scales in layers without LoRA using QZO in the second half.
>
> Based on the results, we report the following findings:
>
> 1. First-order methods consistently outperform zeroth-order methods. This is expected as real gradients are used for optimization rather than approximations. We emphasize that although QLoRA fine-tuning shows low memory usage in memory profiling, this efficiency is achieved only when gradient checkpointing is enabled and a paged optimizer is used. In comparison, QZO and its variant directly achieve memory efficiency through zeroth-order optimization without bells and whistles.
> 2. We highlight that it is possible to combine QZO and QLoRA to improve the performance while keeping the memory efficiency. Based on the results, the QZO+QLoRA variant could generally outperform the original QZO while maintaining a similar low memory footprint. We also emphasize that the QZO+QLoRA variant requires only $8,000$ steps, saving $60\%$ of the total fine-tuning steps required by the original QZO and MeZO. We believe that combining QZO and PEFT methods can be more effectively accomplished than our naive approach, suggesting a promising direction for future research.
>
> |                  | Methods     | Model Precision | Memory Profiling | **SST-2** | **RTE** | **CB** |
> | ---------------- | ----------- | --------------- | ---------------- | --------- | ------- | ------ |
> | **First-order**  | Fine-tuning | 16 bits         | 26.8 GB          | 95.4      | 79.8    | 73.2   |
> |                  | LoRA        | 16 bits         | 14.0 GB          | 95.6      | 83.8    | 71.4   |
> |                  | QLoRA       | 4 bits          | 5.6 GB           | 96.1      | 84.1    | 67.9   |
> | **Zeroth-order** | MeZO        | 16 bits         | 14.8 GB          | 93.0      | 64.6    | 67.9   |
> |                  | QZO         | 4 bits          | 4.8 GB           | 87.6      | 61.7    | 67.9   |
> |                  | QZO+QLoRA   | 4 bits          | 4.9 GB           | 93.3      | 61.7    | 69.6   |

---

> > ### Author Response · Authors · 2025-11-20
> > **Response to Reviewer g1TF - Page 2 /2**
> >
> > > As shown in Table 1, the proposed method suffers from a notable performance drop during training.
> >
> > QZO performs similarly to MeZO, although it has lower performance on some of the tasks. We attribute this phenomenon to the low-precision model used in QZO. Nevertheless, we also highlight that QZO achieves the best memory efficiency among the compared baselines.
> >
> > > Is there a comparison with the Adam optimizer? The gap between the zero-shot optimizer and Adam remains unclear.
> >
> > Thank you for the suggestion! Due to limited computing resources, we are unable to provide full-parameter fine-tuning results with the AdamW optimizer. Nevertheless, you can refer to the LoRA results presented in the rebuttal for Reviewer EThz as an approximate reference. We also think there is a typo in this comment: it may be 'zeroth-order optimizer' rather than 'zero-shot optimizer'.
> >
> > > In addition to the final evaluation results, providing training loss curves would help to more intuitively compare training convergence and stability.
> >
> > Thank you for the suggestion! We have updated the plots of loss curves in Appendix B of our rebuttal revision. The loss curves demonstrate a clear convergence pattern, indicating the training stability of QZO.
> >
> > We thank you again for all the feedback and suggestions, and hope our response could address your concerns.
> >
> >
> >
> > References
> >
> > [1] Hu E J, Shen Y, Wallis P, et al. Lora: Low-rank adaptation of large language models[J]. ICLR, 2022, 1(2): 3.
> >
> > [2] Dettmers T, Pagnoni A, Holtzman A, et al. Qlora: Efficient finetuning of quantized llms[J]. Advances in neural information processing systems, 2023, 36: 10088-10115.

---

> > > ### Comment · Reviewer_g1TF · 2025-11-26
> > >
> > > Thank you for the author's rebuttal. However, it only addresses some of the concerns. One major concern remains regarding the adequacy of the comparisons. The experiments only compare against first-order or zero-order methods, without comparing against commonly used benchmark optimizers like Adam. This means the comparisons are consistently against suboptimal methods, which I believe limits their persuasiveness.

---

> > > > ### Author Response · Authors · 2025-11-28
> > > >
> > > > Thanks for your feedback! Below, we present the experimental results, supplemented with AdamW, to further address your primary concerns. Specifically, we use the paged AdamW-8bit optimizer presented in QLoRA[1] and enable gradient checkpointing, ensuring that the maximum training memory per run does not exceed 50GB on a single NVIDIA RTX A6000. We set the learning rate to $10^{-5}$ with linear decay. Other settings are consistent with the SGD fine-tuning experiments as detailed in our manuscript. We emphasize that, although the memory profiling results indicate acceptable memory usage for AdamW, this is only true when the optimizer states are quantized, and gradient checkpointing is enabled. When the regular AdamW is used with fully sharded data parallel, the memory cost easily exceeds 80 GB (see Figure 1 in our manuscript for more details).
> > > >
> > > > |              |                           | Model Precision | Memory Profiling | SST-2 | RTE  | CB   | BoolQ | SQuAD |
> > > > | ------------ | ------------------------- | --------------- | ---------------- | ----- | ---- | ---- | ----- | ----- |
> > > > | OPT-6.7B     | Fine-tuning w/ AdamW-8bit | 16 bits         | 38.7 GB          | 94.0  | 82.7 | 87.5 | 81.8  | 85.7  |
> > > > |              | Fine-tuning w/ SGD        | 16 bits         | 26.8 GB          | 95.4  | 79.8 | 73.2 | 69.6  | 77.6  |
> > > > |              | Zero-shot                 | 16 bits         | -                | 61.2  | 55.2 | 51.8 | 59.5  | 36.5  |
> > > > |              | Zero-shot-Q               | 4 bits          | -                | 60.1  | 53.8 | 51.8 | 59.1  | 35.9  |
> > > > |              | MeZO                      | 16 bits         | 14.8 GB          | 93.0  | 64.6 | 67.9 | 66.8  | 79.6  |
> > > > |              | QZO                       | 4 bits          | 4.8 GB           | 87.6  | 61.7 | 67.9 | 66.4  | 78.5  |
> > > > |              |                           |                 |                  |       |      |      |       |       |
> > > > | Llama-2-7B   | Fine-tuning w/ AdamW-8bit | 16 bits         | 38.4 GB          | 94.4  | 89.9 | 92.9 | 85.1  | 88.6  |
> > > > |              | Fine-tuning w/ SGD        | 16 bits         | 26.0 GB          | 92.8  | 63.2 | 60.7 | 75.0  | 83.7  |
> > > > |              | Zero-shot                 | 16 bits         | -                | 58.1  | 61.7 | 32.1 | 66.0  | 55.6  |
> > > > |              | Zero-shot-Q               | 4 bits          | -                | 58.5  | 53.4 | 35.7 | 64.6  | 53.6  |
> > > > |              | MeZO                      | 16 bits         | 14.8 GB          | 83.5  | 58.1 | 67.9 | 69.6  | 80.7  |
> > > > |              | QZO                       | 4 bits          | 5.0 GB           | 90.0  | 59.2 | 69.6 | 68.2  | 85.5  |
> > > > |              |                           |                 |                  |       |      |      |       |       |
> > > > | Llama-3.1-8B | Fine-tuning w/ AdamW-8bit | 16 bits         | 45.8 GB          | 95.3  | 90.3 | 96.4 | 87.8  | 90.7  |
> > > > |              | Fine-tuning w/ SGD        | 16 bits         | 31.9 GB          | 93.7  | 71.5 | 62.5 | 83.4  | 84.9  |
> > > > |              | Zero-shot                 | 16 bits         | -                | 59.6  | 45.8 | 46.4 | 66.1  | 64.8  |
> > > > |              | Zero-shot-Q               | 4 bits          | -                | 58.7  | 50.2 | 37.5 | 65.0  | 59.2  |
> > > > |              | MeZO                      | 16 bits         | 20.5 GB          | 92.5  | 70.0 | 91.1 | 83.4  | 86.9  |
> > > > |              | QZO                       | 4 bits          | 6.3 GB           | 93.0  | 66.8 | 69.6 | 78.2  | 88.3  |
> > > >
> > > >
> > > >
> > > > References
> > > >
> > > > [1] Dettmers T, Pagnoni A, Holtzman A, et al. Qlora: Efficient finetuning of quantized llms[J]. Advances in neural information processing systems, 2023, 36: 10088-10115.

---

### Official Review · Reviewer_wVKC · 2025-11-01

**Soundness:** 3
**Presentation:** 4
**Contribution:** 3
**Rating:** 8
**Confidence:** 4

**Summary:**

The paper aims to reduce GPU memory usage for LLMs by employing zeroth-order optimization on quantized neural networks, a method referred to as QZO. The key idea behind QZO is to approximate gradients by perturbing the continuous parameters involved in the quantization process while keeping the model weights quantized. The authors demonstrate that QZO achieves substantial reductions in GPU memory consumption while maintaining strong performance compared to prior methods and baselines that require significantly more memory.

**Strengths:**

- This is a particularly relevant problem, as researchers without access to high-end GPUs often face significant challenges in conducting LLM research due to the models’ large memory requirements.
- The paper is well written, the proposed solution is elegant, and the experiments are carefully designed to evaluate different components of QZO, such as clipping.

**Weaknesses:**

- Using clipping for variance reduction may make the proposed method sensitive to hyperparameter choices. As shown in Figure 3, the clipping parameter has a substantial impact on accuracy, and it is unclear how this parameter should be selected beyond trial and error.

**Questions:**

- In RL, gradient estimates often have high variance since they are computed over trajectories, for instance under the current policy in the on-policy setting. A common approach to reduce variance, which tends to be less brittle than the clipping used in PPO, is to introduce a KL-based regularization term. Could the objective be modified to incorporate such a regularization mechanism instead of relying on clipping?
- While QZO achieves substantial memory reduction, it would be helpful to see plots showing the full trajectory of the loss and test accuracy. How quickly do these approaches converge compared to the baselines?

---

> ### Author Response · Authors · 2025-11-20
> **Response to Reviewer wVKC**
>
> Thank you for the very positive feedback! We appreciate your comments on a significant research topic, good writing, elegant solution, and carefully designed experiments. We post our responses below to further address your concerns.
>
> > Using clipping for variance reduction may make the proposed method sensitive to hyperparameter choices. As shown in Figure 3, the clipping parameter has a substantial impact on accuracy, and it is unclear how this parameter should be selected beyond trial and error.
>
> We clarify that although a quantitative criterion has not been explored, the choice of clipping threshold for the estimated directional derivative (i.e., $C$) is generally flexible. As shown in Figure 3, the impact of $C$ on performance becomes negligible beyond a certain threshold (i.e., $C\ge75$ in the experiments shown in Figure 3). This is reasonable, since clipping in this case corrects the abnormal estimates of directional derivatives, which can cause the training to diverge, while leaving the rest untouched. This also indicates that in practice, one may set $C$ to be a large constant as long as the training does not collapse.
>
> > In RL, gradient estimates often have high variance since they are computed over trajectories, for instance under the current policy in the on-policy setting. A common approach to reduce variance, which tends to be less brittle than the clipping used in PPO, is to introduce a KL-based regularization term. Could the objective be modified to incorporate such a regularization mechanism instead of relying on clipping?
>
> Based on our understanding, the KL-based regularization term in reinforcement learning stabilizes training by penalizing a new policy from being too far from the reference policy, such as the initial SFT model [1,2]. In other words, it is used to reduce discrepancies in the outputs generated by the policy model and its SFT initialization counterparts. As our experiments do not involve reinforcement learning, we are unsure how KL-based regularization will affect gradient estimates. Nevertheless, this technique should be theoretically compatible with directional derivative clipping (DDC) when reinforcement learning is performed, as DDC reduces the variance of gradient estimates, including those computed from the KL regularization term.
>
> > While QZO achieves substantial memory reduction, it would be helpful to see plots showing the full trajectory of the loss and test accuracy. How quickly do these approaches converge compared to the baselines?
>
> Thank you for the suggestion! We have appended plots of loss curves in the updated rebuttal revision of our manuscript for your reference (see Appendix B for more details). When fine-tuning an OPT-6.7B on SST-2 using a single NVIDIA 4090, MeZO training takes approximately 4 hours and 26 minutes, while QZO takes around 2 hours and 16 minutes.
>
> We thank you again for the feedback and hope our responses address your concerns!
>
>
>
> References:
>
> [1] Vieillard N, Kozuno T, Scherrer B, et al. Leverage the average: an analysis of kl regularization in reinforcement learning[J]. Advances in Neural Information Processing Systems, 2020, 33: 12163-12174.
>
> [2] Shao Z, Wang P, Zhu Q, et al. Deepseekmath: Pushing the limits of mathematical reasoning in open language models[J]. arXiv preprint arXiv:2402.03300, 2024.

---

> > ### Comment · Reviewer_wVKC · 2025-11-25
> >
> > Thank you for the response. Figure 4 in Appendix B reports only the training loss for QZO. My question was to compare both the training loss and the test accuracy of QZO with alternatives such as MeZO throughout training, in order to illustrate the convergence behavior of the methods.

---

> > > ### Author Response · Authors · 2025-11-27
> > >
> > > Thank you for the feedback! We have revised Figure 4 to compare QZO and MeZO based on both loss values and test accuracy. The curves show a convergence pattern of QZO similar to its zeroth-order baseline, as well as the training stability. We hope this further addresses your concern!

---

> > > > ### Comment · Reviewer_wVKC · 2025-11-27
> > > >
> > > > Thank you for the plot. I'll maintain my positive rating.

---

### Official Review · Reviewer_EThz · 2025-11-03

**Soundness:** 3
**Presentation:** 4
**Contribution:** 3
**Rating:** 6
**Confidence:** 3

**Summary:**

The paper proposes Quantized Zeroth-Order Optimization (QZO) for fine-tuning quantized neural networks—primarily LLMs—without backpropagation. The key idea is to apply SPSA-style zeroth-order perturbations to the continuous quantization scales while keeping discrete quantized weights fixed, thereby eliminating gradients and optimizer states and drastically reducing memory. A simple Directional Derivative Clipping (DDC) further stabilizes training by reducing gradient-estimate variance.

**Strengths:**

1. Q-SPSA operates on Δ, keeping integer weights fixed; avoids backprop and optimizer states.

2. DDC has theoretical backing and empirical impact (NaNs avoided; improved stability).

3. ~18× memory saving; single-GPU (24 GB) fine-tuning for large models; reduced trainable params/FLOPs vs MeZO.

4. Works across 4-bit (GPTQ) and 2-bit (AQLM) quantization; preliminary coverage of diffusion.

Overall, I think this is a good paper.

**Weaknesses:**

1. The comparison set omits strong practical baselines such as LoRA/QLoRA/AdaLoRA on the same datasets/models/bit-widths. Since QZO and PEFT are orthogonal (QZO tunes scales; PEFT adds low-rank adapters), including them would contextualize accuracy-vs-memory/latency trade-offs more fairly than only MeZO/SGD. (Tables report MeZO and full FT but not PEFT.)

2. The method’s success is acknowledged to depend on PTQ quality, but there’s no systematic study across quantizers (e.g., AWQ/SmoothQuant vs GPTQ/AQLM) at fixed bit-widths to isolate the effect of QZO when quantization error changes.

3. Supervised experiments use small subsampled splits (1k train per task), which can inflate variance and may not reflect real fine-tuning regimes; larger or diverse instruction-tuning sets would strengthen claims.

**Questions:**

1. Can QZO be combined with LoRA/QLoRA to improve accuracy while keeping memory low?

2. How sensitive is QZO to ε and learning rate across bit-widths (2/3/4-bit)? Any principled schedule for C beyond static thresholds?

3. Two forward passes per step suggest similar step-costs to MeZO; what are end-to-end time-to-accuracy comparisons under matched hardware?

---

> ### Author Response · Authors · 2025-11-20
> **Response to Reviewer EThz  - Page 1 / 2**
>
> Thank you for your valuable feedback! We appreciate your positive comments about our method design and experimental results. We make the following responses to address your primary concerns.
>
> > Can QZO be combined with LoRA/QLoRA to improve accuracy while keeping memory low?
>
> Thanks for the suggestion. We first clarify that the proposed QZO is orthogonal to parameter-efficient fine-tuning (PEFT) methods, as it directly tunes the scales of quantized models without requiring any additional trainable adapters. This also indicates that the QZO and PEFT methods can be combined to fine-tune quantized models jointly using the zeroth-order optimizer. For a quantitative study, we conduct a series of experiments to compare the performance of QZO, QLoRA[1], and the combination of both. The results are presented in the following table. Specifically, we use the methods listed in the table to fine-tune a (quantized) OPT-6.7B model. The low-rank adapters are injected into the weight matrices of the `q_proj` and `v_proj` layers across all experiments conducted with PEFT methods. And the LoRA rank and LoRA alpha are set to $8$ and $16$ consistently. We also naively combine QZO with zeroth-order QLoRA to create the variant 'QZO+QLoRA' by fine-tuning low-rank adapters during the first half of training with zeroth-order QLoRA, while training the remaining quantization scales in layers without LoRA using QZO in the second half.
>
> Based on the results, we report the following findings:
>
> 1. First-order methods consistently outperform zeroth-order methods. This is reasonable, since real gradients are used for optimization rather than approximations. We emphasize that although QLoRA fine-tuning shows low memory usage in memory profiling, this efficiency is achieved only when gradient checkpointing is enabled and a paged optimizer is used. In comparison, QZO and its variant directly achieve memory efficiency through zeroth-order optimization without bells and whistles.
> 2. We highlight that it is possible to combine QZO and QLoRA to improve the performance while keeping the memory efficiency. Based on the results, the QZO+QLoRA variant could generally outperform the original QZO while maintaining a similar low memory footprint. We also emphasize that the QZO+QLoRA variant requires only $8,000$ steps, saving $60\%$ of the total fine-tuning steps required by the original QZO and MeZO. We believe that combining QZO and PEFT methods can be more effectively accomplished than our naive approach, suggesting a promising direction for future research.
>
> |                  | Methods     | Model Precision | Memory Profiling | **SST-2** | **RTE** | **CB** |
> | ---------------- | ----------- | --------------- | ---------------- | --------- | ------- | ------ |
> | **First-order**  | Fine-tuning | 16 bits         | 26.8 GB          | 95.4      | 79.8    | 73.2   |
> |                  | LoRA        | 16 bits         | 14.0 GB          | 95.6      | 83.8    | 71.4   |
> |                  | QLoRA       | 4 bits          | 5.6 GB           | 96.1      | 84.1    | 67.9   |
> | **Zeroth-order** | MeZO        | 16 bits         | 14.8 GB          | 93.0      | 64.6    | 67.9   |
> |                  | QZO         | 4 bits          | 4.8 GB           | 87.6      | 61.7    | 67.9   |
> |                  | QZO+QLoRA   | 4 bits          | 4.9 GB           | 93.3      | 61.7    | 69.6   |

---

> > ### Author Response · Authors · 2025-11-20
> > **Response to Reviewer EThz - Page 2 / 2**
> >
> > > How sensitive is QZO to ε and learning rate across bit-widths (2/3/4-bit)? Any principled schedule for C beyond static thresholds?
> >
> > We first emphasize that the perturbation scale $\epsilon$ is a dedicated hyperparameter, as a large $\epsilon$ leads to inaccurate gradient estimates, while an $\epsilon$ too small not only poses a potential risk to numerical stability, but also increases the noise from higher-order terms (note that Eq. 1 and Eq. 5 are derived by first-order Taylor approximation). As per prior practice in zeroth-order optimization[2], we set $\epsilon$ to $10^{-3}$ across all our experiments. Similarly, the learning rate is also fixed to $10^{-7}$ across all the experiments reported. Although the clipping threshold (i.e., $C$) remains static throughout fine-tuning, we clarify that the choice of $C$ is relatively flexible. As shown in Figure 3, $C$ does not have a significant impact on the performance when exceeding a certain threshold (say $75$ in the figure), so one may preset it to a large constant as long as the training does not collapse.
> >
> > > Two forward passes per step suggest similar step-costs to MeZO; what are end-to-end time-to-accuracy comparisons under matched hardware?
> >
> > We acknowledge that QZO also requires two forward passes per iteration, and in our experiments, the total number of optimization steps is the same as those needed by MeZO. Nevertheless, we also clarify that QZO requires less training time than MeZO, as inference kernels can speed up the forward pass for quantized models. For example, when fine-tuning an OPT-6.7B on SST-2 using a single NVIDIA 4090, MeZO training takes approximately 4 hours and 26 minutes, while QZO takes around 2 hours and 16 minutes.
> >
> > We thank you again for the feedback and hope our responses address your concerns!
> >
> >
> >
> > References
> >
> > [1] Dettmers T, Pagnoni A, Holtzman A, et al. Qlora: Efficient finetuning of quantized llms[J]. Advances in neural information processing systems, 2023, 36: 10088-10115.
> >
> > [2] Malladi S, Gao T, Nichani E, et al. Fine-tuning language models with just forward passes[J]. Advances in Neural Information Processing Systems, 2023, 36: 53038-53075.

---

> ### Author Response · Authors · 2025-11-28
>
> Hello! It's been a few days since our initial discussion post. I wonder if our responses have addressed your concerns. If not, please feel free to start another round of discussion. We welcome any valuable suggestions from you to help improve our work! We would also be very grateful if you could raise the final score once your concerns have been addressed. Looking forward to your reply!

---

### Author Response · Authors · 2025-11-29
**Summary of Responses - Page 1 / 2**

We summarize the reviewers' main concerns and our responses below to assist the AC with the final decision during the meta-review.

### Q1. Is there any comparison with PEFT methods? Can QZO be combined with PEFT methods, such as LoRA, to enhance performance? (EThz, g1TF, p1QC)

This response addresses the common concerns of Reviewers EThz, g1TF, and p1QC. We first clarify that the proposed QZO is orthogonal to parameter-efficient fine-tuning (PEFT) methods, as it directly tunes the scales of quantized models without requiring any additional trainable adapters. This also indicates that the QZO and PEFT methods can be combined to fine-tune quantized models jointly using the zeroth-order optimizer. For a quantitative study, we conduct a series of experiments to compare the performance of QZO, QLoRA[1], and the combination of both. The results are presented in the following table. Specifically, we use the methods listed in the table to fine-tune a (quantized) OPT-6.7B model. The low-rank adapters are injected into the weight matrices of the `q_proj` and `v_proj` layers across all experiments conducted with PEFT methods. And the LoRA rank and LoRA alpha are set to $8$ and $16$ consistently. We also naively combine QZO with zeroth-order QLoRA to create the variant 'QZO+QLoRA' by fine-tuning low-rank adapters during the first half of training with zeroth-order QLoRA, while training the remaining quantization scales in layers without LoRA using QZO in the second half.

Based on the results, we report the following findings:

1. First-order methods consistently outperform zeroth-order methods. This is reasonable, since real gradients are used for optimization rather than approximations. We emphasize that although QLoRA fine-tuning shows low memory usage in memory profiling, this efficiency is achieved only when gradient checkpointing is enabled and a paged optimizer is used. In comparison, QZO and its variant directly achieve memory efficiency through zeroth-order optimization without bells and whistles.
2. We highlight that it is possible to combine QZO and QLoRA to improve the performance while keeping the memory efficiency. Based on the results, the QZO+QLoRA variant could generally outperform the original QZO while maintaining a similar low memory footprint. We also emphasize that the QZO+QLoRA variant requires only $8,000$ steps, saving $60\%$ of the total fine-tuning steps required by the original QZO and MeZO[2]. We believe that combining QZO and PEFT methods can be more effectively accomplished than our naive approach, suggesting a promising direction for future research.

|                  | Methods     | Model Precision | Memory Profiling | **SST-2** | **RTE** | **CB** |
| ---------------- | ----------- | --------------- | ---------------- | --------- | ------- | ------ |
| **First-order**  | Fine-tuning | 16 bits         | 26.8 GB          | 95.4      | 79.8    | 73.2   |
|                  | LoRA        | 16 bits         | 14.0 GB          | 95.6      | 83.8    | 71.4   |
|                  | QLoRA       | 4 bits          | 5.6 GB           | 96.1      | 84.1    | 67.9   |
| **Zeroth-order** | MeZO        | 16 bits         | 14.8 GB          | 93.0      | 64.6    | 67.9   |
|                  | QZO         | 4 bits          | 4.8 GB           | 87.6      | 61.7    | 67.9   |
|                  | QZO+QLoRA   | 4 bits          | 4.9 GB           | 93.3      | 61.7    | 69.6   |

### Q2. Is there any plot of loss value and test accuracy to show the convergence behaviour of QZO and its compared baseline? (wVKc, g1TF)

As per the requirements from Reviewer wVKC and g1TF, we have added the loss-accuracy curves in Appendix B in our rebuttal revision to compare the convergence and training stability of QZO and MeZO [2]. The curves show a similar convergence pattern, indicating consistent training stability for QZO, as with its full-precision counterparts.

---

> ### Author Response · Authors · 2025-11-29
> **Summary of Responses - Page 2 / 2**
>
> ### Q3. Some related works are missing (p1QC)
>
> As suggested by Reviewer p1QC, we have added a new paragraph to the 'Related Works' section in our updated rebuttal revision to discuss the relationship between QZO and prior methods [3,4,5]. We highlight two unique advantages of QZO that improve both the efficiency and flexibility in contrast to [3,4,5]: (1) QZO is free from quantization of perturbation noises or re-quantization model weights during each optimization step, and (2) QZO can be adapted to existing post-training quantization methods in a plug-and-play manner. We also apologize that due to a lack of source code for all the proposed methods [3,4,5], we are unable to provide the empirical comparisons during the discussion period.
>
> ### Q4. Is there any theoretical guarantee of convergence for QZO? How long is the training time for QZO compared with MeZO? (mqYe, ETHz)
>
> The convergence rate guarantee is the primary concern for Reviewer mqYe, and Reviewer EThz also raises the issue of training time. We first emphasize that a theoretical study from prior work [2] indicates that the zeroth-order optimizer guarantees a convergence rate similar to SGD, with a slowdown factor proportional to the local effective rank of the Hessian of loss. Although QZO perturbs the quantization scales rather than the model weights in full precision, we believe this finding for the zeroth-order optimizer also supports our method. We also clarify that QZO requires less training time than MeZO [2], as inference kernels can speed up the forward pass for quantized models. For example, when fine-tuning an OPT-6.7B on SST-2 using a single NVIDIA 4090, MeZO training takes approximately 4 hours and 26 minutes, while QZO takes around 2 hours and 16 minutes.
>
> **References**
>
> [1] Dettmers T, Pagnoni A, Holtzman A, et al. Qlora: Efficient finetuning of quantized llms[J]. Advances in neural information processing systems, 2023, 36: 10088-10115.
>
> [2] Malladi S, Gao T, Nichani E, et al. Fine-tuning language models with just forward passes[J]. Advances in Neural Information Processing Systems, 2023, 36: 53038-53075.
>
> [3] Bar N, Giryes R. ZOQO: Zero-Order Quantized Optimization[C]//ICASSP 2025-2025 IEEE International Conference on Acoustics, Speech and Signal Processing (ICASSP). IEEE, 2025: 1-5.
>
> [4] Zhou J, Yang Y, Zhen K, et al. QuZO: Quantized zeroth-order fine-tuning for large language models[J]. arXiv preprint arXiv:2502.12346, 2025.
>
> [5] Feng C, Zhuo S, Zhang X, et al. Stepping forward on the last mile[J]. Advances in Neural Information Processing Systems, 2024, 37: 94851-94870.

---

### Meta-Review · Area_Chair_fihN · 2026-01-08

**Summary:**

Three reviewers support acceptance citing practical utility and elegant execution, while one reviewer raises valid novelty concerns given concurrent work (ZOQO, QuZO, Sparse MeZO). The novelty concerns have merit, but the clear 3-vs-1 majority favors acceptance. As Area Chair, I cannot overturn such a clear reviewer consensus. I recommend acceptance.

**Reviewer Concerns:**

see above

**Reviewer Scores:**

Discussion was sufficient; the novelty disagreement reflects genuine differences in how reviewers weigh practical impact versus originality, and scores would have remained similar.

---

### Decision · Program_Chairs · 2026-01-26

Accept (Poster)